# Genome-scale RNA interference profiling of *Trypanosoma brucei* cell cycle progression defects

Catarina A. Marques[1,3,4], Melanie Ridgway [1,4], Michele Tinti [1,4], Andrew Cassidy[2] & David Horn [1] ✉

Trypanosomatids, which include major pathogens of humans and livestock, are flagellated protozoa for which cell cycle controls and the underlying mechanisms are not completely understood. Here, we describe a genome-wide RNA-interference library screen for cell cycle defects in *Trypanosoma brucei*. We induced massive parallel knockdown, sorted the perturbed population using high-throughput flow cytometry, deep-sequenced RNAi-targets from each stage and digitally reconstructed cell cycle profiles at a genomic scale; also enabling data visualisation using an online tool (https://tryp-cycle.pages.dev/). Analysis of several hundred genes that impact cell cycle progression reveals >100 flagellar component knockdowns linked to genome endoreduplication, evidence for metabolic control of the $G_1$-S transition, surface antigen regulatory mRNA-binding protein knockdowns linked to $G_2M$ accumulation, and a putative nucleoredoxin required for both mitochondrial genome segregation and for mitosis. The outputs provide comprehensive functional genomic evidence for the known and novel machineries, pathways and regulators that coordinate trypanosome cell cycle progression.

The canonical eukaryotic cell cycle encompasses discrete phases: $G_1$ (gap 1), when the cell prepares for DNA replication; S (synthesis) phase, when nuclear DNA replication takes place; $G_2$ (gap 2), when the cell prepares for mitosis; and M (mitosis) when the replicated DNA is segregated and the nucleus divides[1]. Mitosis is followed by cytokinesis (cell division), generating two daughter cells[2]. Rate-limiting mechanisms that facilitate quality control are relieved at discrete points. Thus, anomalies occurring during cell cycle progression can result in cell cycle delay or arrest, to allow the cell to resolve the anomaly; in cell death, if the anomaly cannot be resolved or, among other outcomes, in carcinogenesis. Therefore, progression through the cell cycle is typically under strict checkpoint control; the $G_1$-S, intra S phase, $G_2$-M and spindle checkpoints control the onset of S phase, S phase progression, the onset of M phase and M phase progression, respectively[3]. These

processes have been extensively studied, particularly because cell cycle defects are common triggers for carcinogenesis[4]. However, our understanding of the evolution and mechanisms of eukaryotic cell cycle progression control derives primarily from studies on the opisthokonts (including animals and fungi), with relatively fewer studies on divergent eukaryotes, such as the trypanosomatids[1].

The trypanosomatids are flagellated protozoa and include parasites that cause a range of neglected tropical diseases that have major impacts on human and animal health. The African trypanosome, *Trypanosoma brucei*, is transmitted by tsetse flies and causes both human and animal diseases, sleeping sickness and nagana, respectively, across sub-Saharan Africa[5]. *T. brucei* has emerged as a highly tractable experimental system, both as a parasite and as a model organism[6]. For example, the *T. brucei* flagellum[7] serves as a model for studies on

[1]Wellcome Centre for Anti-Infectives Research, School of Life Sciences, University of Dundee, Dow Street, Dundee DD1 5EH, UK. [2]Tayside Centre for Genomic Analysis, Ninewells Hospital and School of Medicine, Dundee DD1 9SY, UK. [3]Present address: Wellcome Centre for Integrative Parasitology, University of Glasgow, 120 University Place, Glasgow G12 8TA, UK. [4]These authors contributed equally: Catarina A. Marques, Melanie Ridgway, Michele Tinti. ✉ e-mail: d.horn@dundee.ac.uk

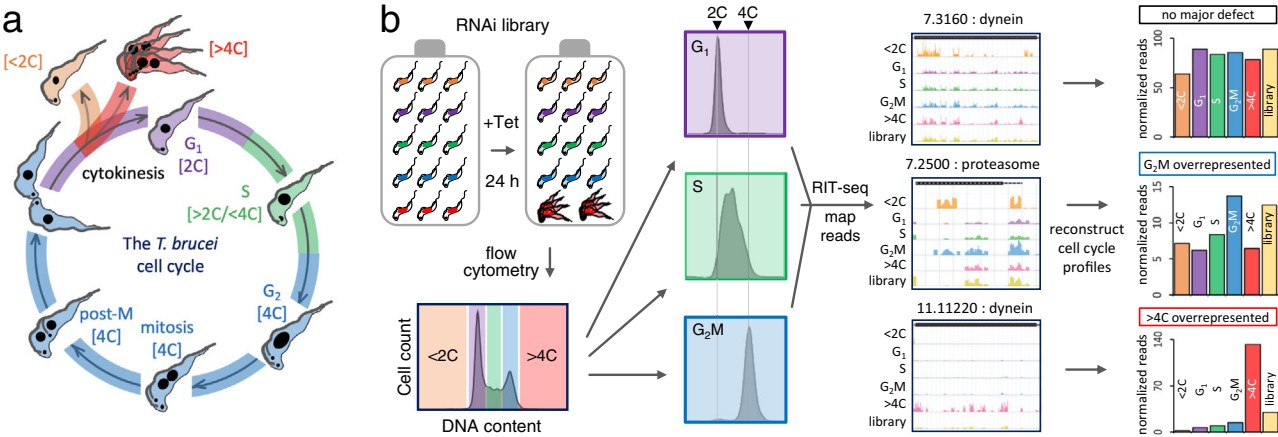

**Fig. 1 | A genome-wide conditional knockdown screen for cell cycle progression defects. a** Schematic representation of the bloodstream form *T. brucei* cell cycle, also showing aberrant sub-2C and >4 C (multiple nuclei) phenotypes; anuclear zoids are not shown as they are undetectable by RIT-seq. **b** The schematic illustrates the RIT-seq screen; massive parallel induction of RNAi with tetracycline (Tet), followed by flow cytometry and RIT-seq, allowing for reconstruction of cell cycle profiles, using mapped reads from each knockdown. Each read-mapping profile encompasses the gene of interest and associated untranslated regions present in the cognate mRNA. The library data represents the uninduced and unsorted population. GeneIDs, Tb927.7.3160 for example, are indicated without the common 'Tb927.' component.

human ciliopathies[8–11]. Divergent features, shared with other pathogenic trypanosomatids, such as *Trypanosoma cruzi* and *Leishmania*, include glycolysis compartmentalised within glycosomes[12], a single mitochondrion with a complex mitochondrial DNA structure known as the kinetoplast[13] and polycistronic transcription of almost every gene[14]. Widespread, and constitutive, polycistronic transcription in trypanosomatids places major emphasis on post-transcriptional controls by mRNA binding proteins (RBPs) and post-translational controls, involving protein phosphorylation, for example.

Studies that focus on cell cycle controls in *T. brucei* have revealed features conserved with other well-studied eukaryotes, but also divergent features[13,15]. Notably, the available evidence suggests that certain cell cycle checkpoints are absent. For example, cytokinesis can occur independent of either mitosis or nuclear DNA synthesis in the insect stage of *T. brucei*[16]. Moreover, functions previously thought to be fulfilled by highly conserved proteins employ lineage-specific or highly divergent proteins in trypanosomatids. The kinetochore complex, which directs chromosome segregation, is trypanosomatid-specific, for example[17], while the origin recognition complex (ORC), involved in DNA replication initiation, is highly divergent[18]. In terms of high-throughput studies, transcriptome[19] and proteome[20] monitoring during the *T. brucei* cell cycle revealed hundreds of regulated mRNAs and proteins, while phosphoproteomic analysis revealed dynamic phosphorylation of several RBPs[21]. Divergence presents substantial outstanding challenges, however, since many *T. brucei* genes have not yet been assigned a specific function, and many cell cycle regulators likely remain to be identified. High-throughput, genome-scale functional genetic screens can be used to simultaneously assess every gene in a genome for a role in a particular process. We developed RNA Interference Target sequencing (RIT-seq) for *T. brucei*[22] and previously generated genome-scale fitness profiles, facilitating essentiality predictions and the prioritisation of potential drug targets[23].

Here, we describe a genome-scale RIT-seq screen to identify cell cycle controls and regulators in bloodstream form African trypanosomes. Following induction of knockdown, cells were sorted according to their DNA content using fluorescence-activated cell sorting (FACS). The sorted populations were the G₁, S and G₂M cell cycle stages as well as perturbed cell populations with either less DNA than typically found in G₁ or more DNA than typically found in G₂M. RIT-seq analysis was carried out for each sorted population and cell cycle profiles were digitally reconstructed for each knockdown using sequencing read-counts. This genome-wide screen reveals the protein complexes, pathways and signalling factors required for progressive steps through the trypanosome cell cycle. For example, glycolytic enzymes are shown to be required for G₁/S progression, CMG (Cdc45-MCM-GINS) complex components are shown to be required for DNA replication, proteasome and kinetochore complex components are shown to be required for G₂M progression, and flagellar components as well as cytokinesis initiation factors are shown to be required for cytokinesis. Two hits were selected for further validation, one of which is shown to be a cell cycle regulated putative nucleoredoxin involved in coordinating segregation of the mitochondrial kinetoplast and the nuclear genome.

## Results

### A genome-wide conditional knockdown screen for cell cycle progression defects

Bloodstream form *T. brucei* are readily grown in cell culture, with exponential proliferation and a doubling time of approximately 6.5 h. The *T. brucei* nuclear genome is typically diploid such that G₁ cells have a 2 C genome content; C represents the haploid DNA content. Cells progressing through nuclear S phase, and replicating their DNA, have a genome content between 2 C and 4 C, while cells that have completed DNA replication (G₂M) have a 4 C DNA content (Fig. 1a). Mitosis and cytokinesis then produce two daughter cells with a 2 C DNA content. Some perturbations yield defects involving a 'short-circuit', whereby S phase, mitosis and/or cytokinesis are skipped, producing sub-2C cells or over-replicated, polyploid (>4 C) cells (Fig. 1a). Polyploid cells arise due to endoreduplication, additional rounds of DNA replication without cytokinesis, either with[24] or without[25,26] mitosis, yielding cells with multiple nuclei or with polyploid nuclei, respectively.

We devised a high-throughput RNA interference (RNAi) target sequencing (RIT-seq) screen to identify cell cycle controls and regulators at a genomic scale. Key features of RIT-seq screening include: first, use of a high-complexity *T. brucei* RNAi library comprising, in this case, approximately one million clones; second, massive parallel tetracycline-inducible expression of cognate dsRNA; and third, deep sequencing, mapping and counting of mapped reads derived from RNAi target fragments[22]. Each clone in the library has one of approximately 100,000 different RNAi target fragments (250–1500 bp) between head-to-head inducible T7-phage promoters. This is achieved by targeting each cassette to a specific, single chromosomal locus that supports robust and reproducible inducible expression[22,27]. Inducibly expressed long dsRNA is then processed to siRNA by the native RNAi

machinery[28]. Complexity and depth of genome coverage in the library are critical, in that similar phenotypes produced by multiple clones with distinct RNAi target fragments against the same gene provide cross-validation. Improvements in reference genome annotation[29], next generation sequencing technology and sequence data analysis tools (see Methods) have also greatly facilitated quantitative phenotypic analysis using short-read sequence data.

Briefly, we induced massive parallel knockdown in an asynchronous *T. brucei* bloodstream form RNAi library for 24 h, fixed the cells, stained their DNA with propidium iodide (PI) and then used fluorescence-activated cell sorting (FACS) to divide the perturbed cell population into; sub-diploid (<2 C), G$_1$ (2 C), S (between 2 C and 4 C), G$_2$M (4 C) and over-replicated (>4 C) pools (Supplementary Fig. 1). Fixation and staining with the fluorescent DNA intercalating dye were pre-optimised for high-throughput sorting (see Materials and Methods). Approximately 10 million cells were collected for each of the G$_1$, S and G$_2$M pools and samples from these pools were checked post-sorting to assess their purity (Fig. 1b, Supplementary Fig. 1). For the perturbed and less abundant <2 C and >4 C pools, less than one million cells were collected; these pools were retained in their entirety for RIT-seq analysis.

RIT-seq was carried out for both the uninduced and induced, unsorted library controls, and for each of the five induced and sorted pools of cells as described in the Methods section. Briefly, we extracted genomic DNA from each sample, amplified DNA fragments containing each RNAi target fragment in PCR reactions (Supplementary Fig. 1) and used the amplified products to generate Illumina sequencing libraries. Analysis of sequencing reads mapped to the reference genome yielded counts for both total reads as well as reads containing the barcode (GTGAGGCCTCGCGA) that flanks each RNAi target fragment; the presence of the barcode confirmed that reads were derived from a specific RNAi target fragment and not from elsewhere in the genome. We derived counts of reads mapped to each of >7200 non-redundant gene sequences in the uninduced and induced, unsorted library controls and in each of the five sorted samples. We selected the 24 h timepoint, equivalent to approximately 3.5 population doubling times, for the current analysis. We found that reads for 23.4% of genes were diminished by >3-fold following 72 h of knockdown in our prior RIT-seq study[23], while reads for only 0.6% of genes dropped by >3-fold following 24 h of knockdown in the unsorted control samples analysed here (see Supplementary Fig. 2a, b). Thus, 24 h should have allowed sufficient time for the development of robust inducible phenotypes and also captured perturbed cells before they were critically diminished due to loss-of-fitness. An unanticipated feature that emerged from this analysis of prior RIT-seq data was that knockdown of proteins associated with DNA replication typically failed to register a major loss-of-fitness (Supplementary Fig. 2a, b). This suggested that a reduced rate of DNA replication can be tolerated, albeit extending S phase (see below) but having relatively little impact on viability. Each sorted sample library yielded between 23 and 37 million mapped read-pairs; <2 C = 37 M, G$_1$ = 35 M, S = 30 M, G$_2$M = 23 M, >4 C = 25 M; this set of five samples yielded data for >7000 genes which equates to >35,000 RNAi data-points (Supplementary data 1).

The RIT-seq digital data for individual genes following knockdown provided a measure of abundance in each pool and were, therefore, used to digitally reconstruct cell cycle profiles for individual gene knockdowns (Fig. 1b). We expected to observe accumulation of particular knockdowns in specific cell cycle phase pools, thereby reflecting specific defects. This was indeed the case, and some examples are shown to illustrate; no major defect, G$_2$M overrepresented or >4 C overrepresented, following knockdown (Fig. 1b). These outputs suggest that loss of a cytoplasmic dynein heavy chain (7.3160) does not perturb cell cycle distribution; that the proteasome is required to complete G$_2$M (see below); and that knockdown of a flagellar axonemal dynein heavy chain (11.11220) results in endoreduplication in the

absence of cytokinesis; dyneins are cytoskeletal motor proteins that either move along microtubules or drive microtubule sliding, to produce a flagellar beat, for example[30].

## Validation and identification of genes linked to cell cycle defects

The *T. brucei* core genome comprises a non-redundant set of over 7200 protein-coding sequences, for which we were now able to digitally reconstruct cell cycle profiles following knockdown. First, we examined knockdowns reporting an overrepresentation of >4 C cells, indicating endoreduplication, which yielded 284 genes (Fig. 2a, left-hand panel; Supplementary data 1). The >4 C phenotype was previously observed following α-tubulin knockdown in a landmark study that first described RNAi in *T. brucei*[24] and, indeed, we observed pronounced overrepresentation of >4 C cells for both adjacent α-tubulin and β-tubulin gene knockdowns (Fig. 2a, middle and right-hand panel). We then examined knockdowns reporting an overrepresentation of <2 C cells, indicating a reduced DNA content, which yielded 10 hits (Fig. 2b, left-hand panel; Supplementary data 1). Haploid cells were previously observed following DOT1A knockdown[31] and, consistent with the previous report, we observed pronounced overrepresentation of <2 C cells for the DOT1A gene knockdown (Fig. 2b, middle and right-hand panel); we are not aware of other knockdowns reported to yield a similar phenotype. Indeed, other '<2 C hits' mostly encode small hypothetical proteins, seven of which are 73 ± 11% shorter than the average, consistent with low read-count and under-sampling for these hits (Supplementary Fig. 2c). The remaining two hits are a histone chaperone (ASF1B) and a glycolytic enzyme (PFK). Together, these results provided initial validation for the >4 C and <2 C components of the screen.

Next, we turned our attention to knockdowns reporting an overrepresentation of G$_1$, S phase or G$_2$M cells. The pools of knockdowns that registered >25% overrepresented read counts in each of these categories are highlighted in the RadViz plot in Fig. 2c (also see Supplementary data 1) and data for an example from each category are shown in Fig. 2d; the glycolytic enzyme, aldolase, reported 104% increase in G$_1$ cells (further details below); the proliferating cell nuclear antigen (PCNA), a DNA sliding clamp that is a central component of the replication machinery[32], reported 25% increase in S phase cells and 13% increase in G$_2$M cells, consistent with prior analysis[33]; and PrimPol-like 2 (PPL2), a post-replication translesion polymerase, reported 65% increase in G$_2$M cells, also consistent with prior analysis[34]. These results provided initial validation for the G$_1$, S phase and G$_2$M components of the screen. The full dataset can be searched and browsed using an interactive, open access, online data visualization tool (see Supplementary Fig. 3; https://tryp-cycle.pages.dev/).

Overall, the five components of the screen yielded 1198 genes that registered a cell cycle defect, based on the thresholds applied above. This is 16.6% of the 7205 genes analysed, and the distributions of these genes among the five arms of the screen are shown in the Venn diagram in Fig. 2e. Since we predicted that knockdowns associated with a cell cycle defect were more likely to also register a growth defect, we compared these datasets to prior RIT-seq fitness profiling data[23]. All groups of genes that registered cell cycle defects, except for the small <2 C set, were significantly enriched for genes that previously registered a loss-of-fitness phenotype following knockdown in bloodstream form cells ($\chi^2$ test; <2 C, $p = 0.93$; G$_1$, $p = 0.04$; S phase, $p = 1.3^{-4}$; G$_2$M, $p = 1.3^{-23}$; >4 C, $p = 9.4^{-213}$), consistent with loss-of-fitness as a common outcome following a cell cycle progression defect. Taken together, the analyses above provided validation for the RIT-seq based cell cycle phenotyping approach and yielded >1000 candidate proteins that impact progression through specific steps of the *T. brucei* cell cycle.

## Cytokinesis defects associated with endoreduplication

In bloodstream form *T. brucei*, defective >4 C cells can arise due to endoreduplication without cytokinesis, either with[24] or without[25,26]

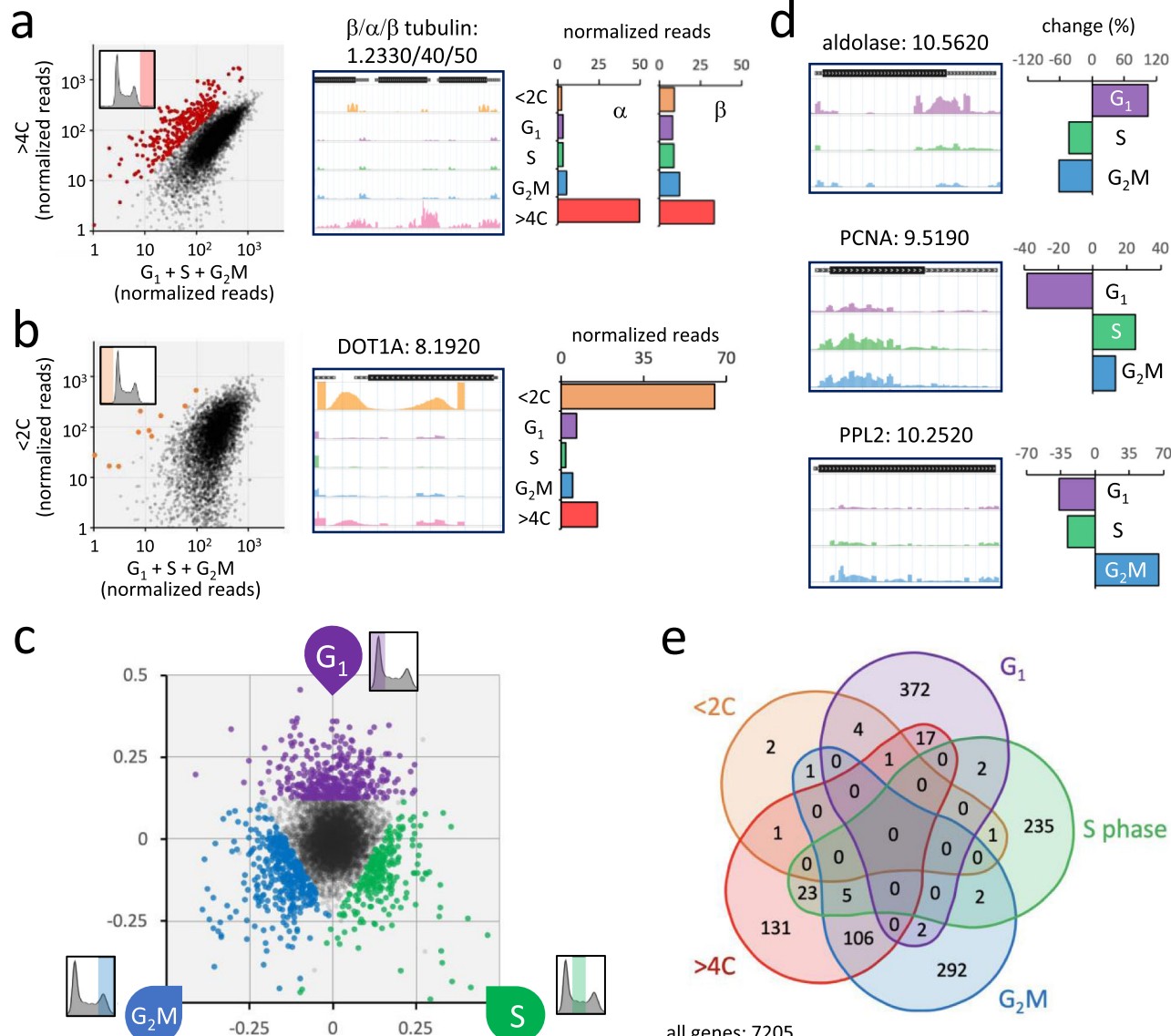

**Fig. 2 | Validation and identification of >1000 candidates linked to cell cycle defects. a** The plot on the left shows knockdowns overrepresented in the >4 C experiment in red; those with reads in the >4 C pool that exceeded the mean fold-change value by >1.75 times the SD, equivalent to >1.117-fold the sum of reads in the G₁, S phase and G₂M samples combined. The read-mapping profile and read-counts for α/β-tubulin are shown to the right. **b** The plot on the left shows knockdowns overrepresented in the sub-2C experiment in orange; those with reads in the sub-2C pool that exceeded the mean fold-change value by >1.75 times the SD, equivalent to

>4-fold the sum of reads in the G₁, S phase and G₂M samples combined. The read-mapping profile and read-counts for DOT1A are shown to the right. **c** The RadViz plot shows knockdowns that registered >25% overrepresented read-counts in the G₁ (purple), S phase (green), or G₂M (blue) categories. **d** Read-mapping profiles and relative read-counts for example hits. PCNA, proliferating cell nuclear antigen; PPL2, PrimPol-like 2. **e** The Venn diagram shows the distribution of knockdowns overrepresented in each arm of the screen.

mitosis. Endoreduplication defects were previously observed following knockdown of α-tubulin[24] or flagellar proteins[7,35]; consistent with the view that flagellar beat is required for cytokinesis in bloodstream form *T. brucei*. As shown above, dynein heavy chain (see Fig. 1b), α-tubulin and β-tubulin (see Fig. 2a) knockdowns were amongst 284 knockdowns overrepresented in the endoreduplicated pool in our screen. Gene Ontology (GO) annotations, which provide structured descriptions of gene products in terms of functions, processes, and compartments, were assessed to further profile this cohort of knockdowns. Terms overrepresented in association with an endoreduplication defect included 'dynein', 'intraflagellar transport' (IFT), 'axoneme' and 'cytoskeleton', and also 'chaperonin T-complex', 'cytokinesis' and 'cell cycle' (Fig. 3a). The violin plot in Fig. 3b shows specific enrichment of IFT and dynein knockdowns in association with endoreduplication. Exocyst components, primarily involved in exocytosis[36], were included

as a control cohort since none of the exocyst components registered enrichment in the >4 C pool, nor in any other experimental pool analysed here (see below). Enrichment of individual chaperonin T-complex components, dyneins, and IFT factors in the >4 C pool is illustrated in Fig. 3c. The chaperonin T-complex is involved in tubulin and actin folding[37] and, notably, actin knockdown was also associated with endoreduplication (Supplementary Fig. 4).

The heat-map in Fig. 3d shows the data for all five sorted pools for the cohorts described above and for additional cohorts of knockdowns enriched in the >4 C pool; these include additional dynein chains, radial spoke proteins, extra-axonemal paraflagellar rod (PFR) proteins, as well as nucleoporins. The gallery in Fig. 3e shows examples of RIT-seq read-mapping profiles for twenty-six individual genes that register >4 C enrichment following knockdown. In addition to the categories above, these include the inner arm dynein 5-1[38], FAZ

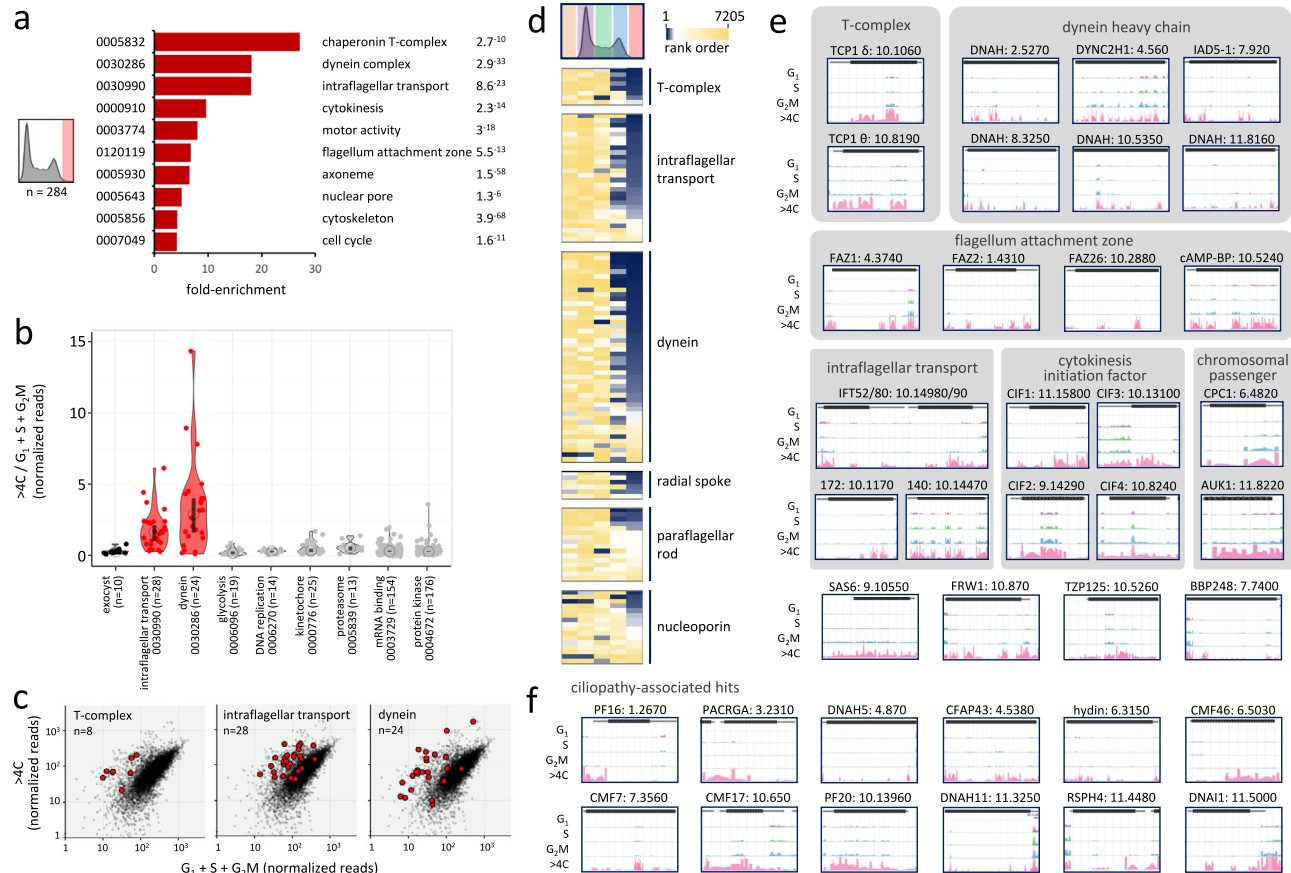

**Fig. 3 | Cytokinesis defects associated with endoreduplication. a** The bar-graph shows enriched Gene Ontology terms associated with the >4 C hits, those that exceed the mean fold-change value in this set by >1.75 times the SD. *P*-values are shown on the right. **b** The violin plot shows relative >4 C read-counts for cohorts of genes and reflects data distribution. Open circles indicate median values and the vertical bars indicate 95% confidence intervals. Significantly overrepresented cohorts are indicated in red. **c** The plots show overrepresentation of T-complex, intraflagellar transport (IFT) factors in red in the >4 C experiment. **d** The heatmaps show relative representation in all five sorted pools for the above and additional cohorts of knockdowns; blue, most overrepresented. **e** Example read-mapping profiles for hits overrepresented in the >4 C pool. **f** Example read-mapping profiles for ciliopathy-associated hits overrepresented in the >4 C pool. CMF Component of Motile Flagella, CFAP Cilia and Flagella Associated Protein.

proteins which mediate attachment of the flagellum to the cell body[39]; all four cytokinesis initiation factors CIF1-4[40], and chromosomal passenger complex components, including CPC1 and the aurora B kinase, AUK1. AUK1 and CPC1 are spindle-associated and regulate mitosis and cytokinesis[26,41]. Notably, endoreduplication was reported previously following AUK1 knockdown in bloodstream form *T. brucei*[42] and this is the kinase with the most pronounced overrepresentation in our >4 C dataset. The next >4 C overrepresented kinase is the CMGC/RCK (Tb927.3.690), knockdown of which previously yielded a striking cytokinesis defect[43].

Additional examples of genes registering >4 C overrepresentation include the centriole cartwheel protein SAS6[44], the cleavage furrow-localizing protein FRW1[45], the basal body−axoneme transition zone protein TZP125[46] and the basal body protein BBP248[47]. One hundred additional examples are shown in Supplementary Fig. 4, including intermediate and light chain dyneins, other flagellum-associated factors, radial spoke proteins, components of motile flagella, flagellum attachment and transition zone proteins, kinesins[48,49], nucleoporins[50], and many previously uncharacterised hypothetical proteins. Some other notable examples include the microtubule-severing katanin KAT80[51], the dynein regulatory factor trypanin[52], the AIR9 microtubule associated protein[53], CAP51V[54] and importin, IMP1[55].

Orthologues of several *T. brucei* flagellar proteins have previously been linked to debilitating human ciliopathies, such that the trypanosome flagellum is exploited as a model for studies on these defects[7].

Defects in intraflagellar dynein transport are associated with respiratory infections, for example[9]. Orthologues of DNAH (10.5350 and 11.8160, Fig. 3e) are linked to male infertility, while additional ciliopathy-associated orthologues which register overrepresentation in the >4 C pool are shown in Fig. 3f and Supplementary Fig. 4. These include orthologues of proteins linked to primary ciliary dyskinesia (DNAH5, DNAH11, RSPH4 and DNAI1)[11]; male infertility (PF16, PACRGA, CFAP43 and CMF7/TbCFAP44)[7,10]; and cone-rod dystrophies, as well as other ocular defects (CMF17, CMF39 and CMF46)[8].

From analysis of knockdowns overrepresented in the >4 C pool, we conclude that RIT-seq screening provided comprehensive genome-scale identification of cytokinesis defects in bloodstream form *T. brucei*. Endoreduplication appears to be a common outcome following a cytokinesis defect. Amongst hundreds of genes required for progression through cytokinesis, flagellar proteins featured prominently, including the majority of dynein chains and intraflagellar transport factors. Many of these factors are essential for viability and include potential druggable targets in trypanosomatids, as well as orthologues of proteins associated with ciliopathies.

## Defects producing sub-diploid cells

A DNA replication or mitosis defect followed by cytokinesis may result in generation of cells that retain nuclear DNA with a sub-2C DNA content. We emphasise retention of nuclear DNA here because *T.*

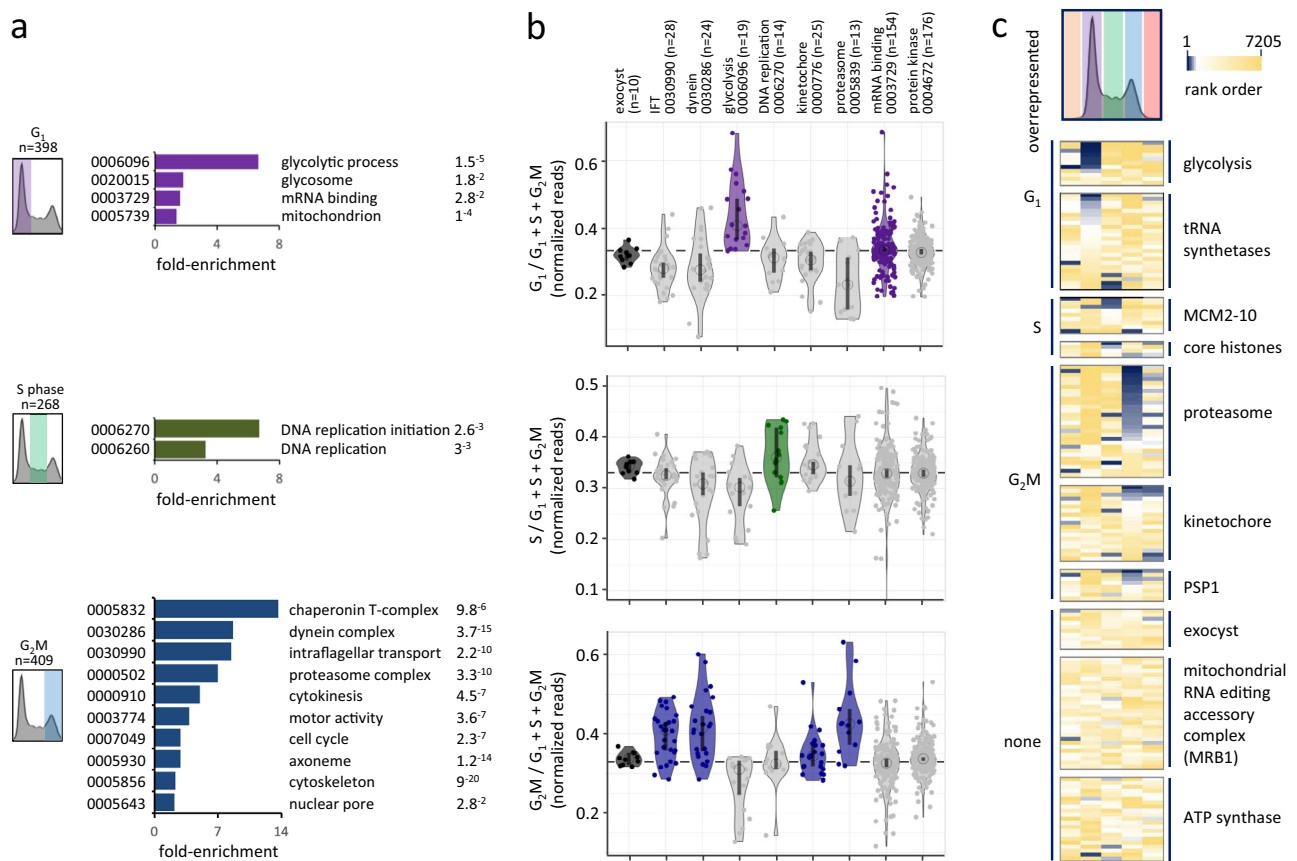

**Fig. 4 | A profile of $G_1$, S phase and $G_2M$ defects. a** The bar-graphs show enriched Gene Ontology terms associated with the $G_1$, S phase or $G_2M$ hits, those that register >25% overrepresented read-count in each of these categories. $P$-values are shown on the right. **b** The violin plots show relative $G_1$, S phase or $G_2M$ read-counts for cohorts of genes and reflect data distribution. Open circles indicate median values and the vertical bars indicate 95% confidence intervals. Overrepresented cohorts are indicated in purple, green and blue, respectively. IFT intraflagellar transport. **c** The heatmaps show relative representation in all five sorted pools for the above and additional cohorts of knockdowns; blue, most overrepresented. MCM mini-chromosome maintenance, PSP1 DNA polymerase suppressor 1.

*brucei* cells lacking nuclear DNA, referred to as zoids, have been reported previously as a result of asymmetrical cell division. Zoids are typically observed when DNA replication or mitosis are perturbed in insect stage cells[16,25,56]. The zoid phenotype is typically either absent or less pronounced in the developmentally distinct bloodstream form cells[57] that we analysed here. Nevertheless, any zoids present in the <2 C pool will not have been detected using RIT-seq, since detection relies upon the presence of a nuclear RNAi target fragment.

Ten knockdowns were overrepresented in the <2 C RIT-seq screening dataset (Supplementary data 1), including the previously identified histone methyltransferase, DOT1A (Fig. 2b). DOT1A is responsible for dimethylation of histone H3K76, and DOT1A knockdown results in mitosis and cytokinesis without DNA replication, generating cells with a haploid DNA content [31]. Our data suggest that few additional knockdowns yield a similar phenotype in bloodstream form *T. brucei*.

## A profile of $G_1$, S phase and $G_2M$ defects

We next analysed knockdowns overrepresented in the $G_1$, S phase or $G_2M$ pools. Several hundred knockdowns registered >25% overrepresented read counts in each of these categories (Fig. 2c, e). GO annotations within each cohort revealed a number of enriched terms (Fig. 4a). Overrepresented knockdowns were associated with glycolysis, mRNA binding and the mitochondrion in the $G_1$ pool, with DNA replication in the S phase pool and with a broadly similar profile to that seen for the >4 C set in the $G_2M$ pool.

The violin plots in Fig. 4b show specific enrichment of individual knockdowns for glycolytic enzymes and a subset of mRNA binding

proteins in the $G_1$ pool, for DNA replication factors in the S phase pool, and proteasome components and a subset of kinetochore components in the $G_2M$ pool (Fig. 4b). Overlap between knockdowns that accumulate in both the $G_2M$ and >4 C pools likely reflects cytokinesis defects with cells accumulating both before and after endoreduplication; compare $G_2M$ and >4 C data for IFT factors and dyneins in Fig. 4b and Fig. 3b, for example. Other mitosis or cytokinesis-perturbed phenotypes are likely not associated with substantial endoreduplication; see the kinetochore and proteasome cohorts in Fig. 4b, for example. Once again, the exocyst provided a control cohort with no components registering enrichment in the $G_1$, S phase or $G_2M$ pools following knockdown (Fig. 4b).

The heat-map in Fig. 4c shows the data for all five sorted pools for the cohorts described above and for additional knockdowns enriched in the $G_1$ or S phase (tRNA synthetases), S phase (core histones) or $G_2M$ pools (PSP1, DNA polymerase suppressor 1), or not enriched in any pool. These latter sets provide further controls that do not appear to have substantial impacts on cell cycle progression, including the mitochondrial RNA editing accessory complex MRB1[58] and the mitochondrial ATP synthase complex V[59]. Thus, we identify protein complexes, pathways and regulatory factors that are specifically required for progressive steps through the trypanosome cell cycle.

## Pathways and protein complexes associated with $G_1$, S phase and $G_2M$ defects

We next explored some of the cohorts of hits described above in more detail. Glycolytic enzymes are particularly prominent amongst

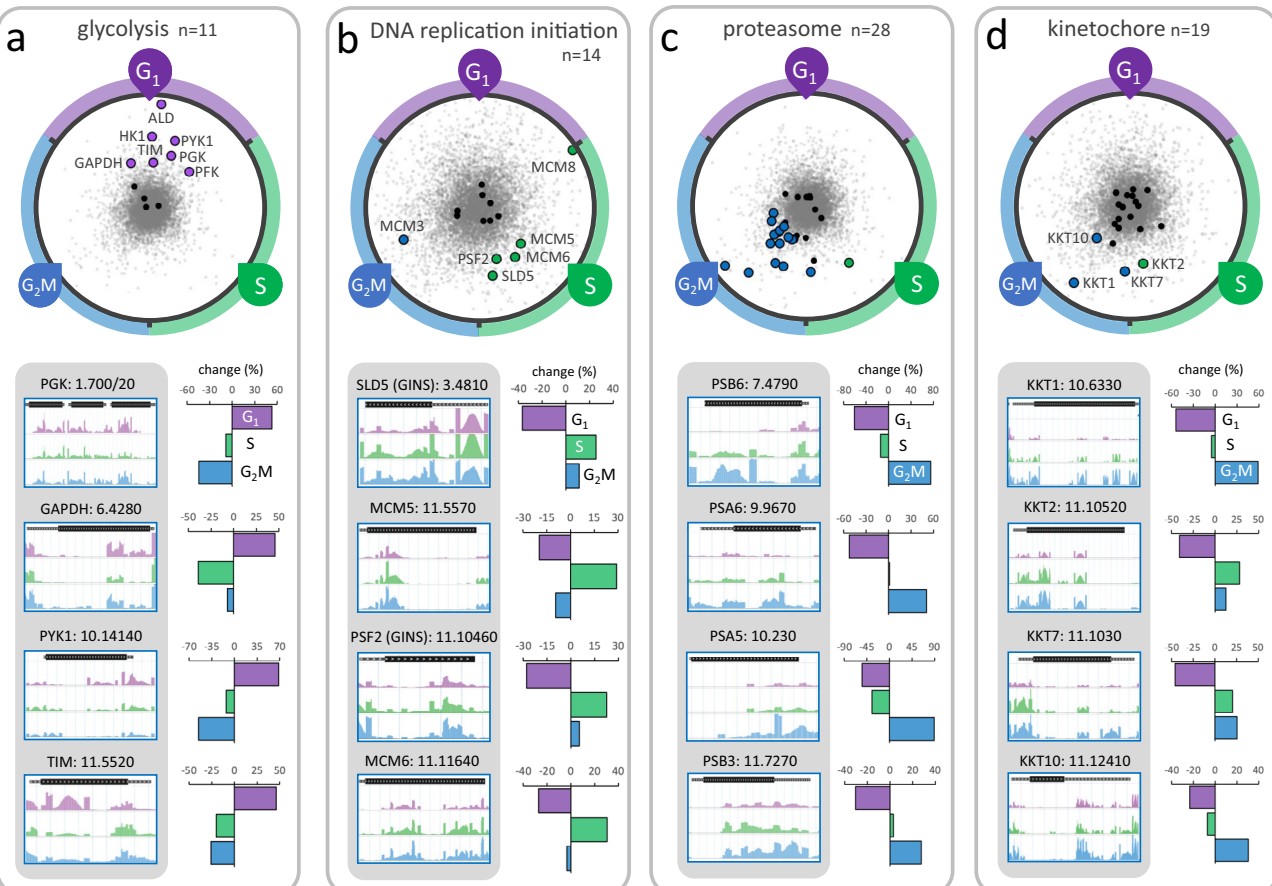

**Fig. 5 | Protein complexes and pathways associated with $G_1$, S phase and $G_2M$ defects. a** The RadViz plot shows glycolytic enzyme knockdowns. Those that registered >25% overrepresented read-counts in the $G_1$ category are indicated. Black data-points indicate other genes from each cohort. Grey data-points indicate all other genes. The read-mapping profiles and relative read-counts in the lower panel show example hits. **b** As in **a** but for DNA replication initiation factor knockdowns that registered >25% overrepresented read-counts, primarily in the S phase category. **c** As in **a** but for proteasome component knockdowns that registered >25% overrepresented read-counts, primarily in the $G_2M$ category. **d** As in **a** but for kinetochore component knockdowns that registered >25% overrepresented read-counts, primarily in the $G_2M$ category.

knockdowns that accumulate in the $G_1$ pool, and we illustrate the RIT-seq profiling data for these enzymes in Fig. 5a. Seven of eleven glycolytic enzyme knockdowns register >25% overrepresentation in the $G_1$ pool; hexokinase, phosphofructokinase, aldolase (see Fig. 2c), triose-phosphate isomerase, glyceraldehyde 3-phosphate dehydrogenase, phosphoglycerate kinase C and pyruvate kinase. Glycolysis operates in peroxisome-like organelles known as glycosomes in trypanosomes and is thought to be the single source of ATP in bloodstream form cells[12]. Glycolysis also provides metabolic intermediates that support nucleotide production. Notably, mammalian cell proliferation is accompanied by activation of glycolysis, and the Warburg effect relates to this phenomenon in oncology[60,61]. Indeed, hexokinase regulates the $G_1/S$ checkpoint in tumour cells[62]. The results are also consistent with the observation that *T. brucei* accumulate in $G_1$ or $G_0$ under growth-limiting conditions[63] or during differentiation to the non-dividing stumpy form[64], possibly reflecting a role for glucose sensing in differentiation[65]. Notably, glycolytic enzymes are downregulated 6.7 +/−5.2-fold in stumpy-form cells[66]. We conclude that, as in other organisms[67], there is metabolic control of the cell cycle and a nutrient sensitive restriction point in *T. brucei*, with glycolysis playing a role in the $G_1$ to S phase transition and possibly also the $G_1/G_0$ transition.

DNA replication initiation factors are particularly prominent amongst knockdowns that accumulate in S phase and we illustrate the RIT-seq profiling data for these factors in Fig. 5b. Five knockdowns that register >25% overrepresentation in the S phase pool are components of the eukaryotic replicative helicase, the CMG (Cdc45-MCM-GINS)

complex. At the core of this complex is the minichromosome maintenance complex (MCM2-7), a helicase that unwinds the duplex DNA ahead of the moving replication fork[68]. Identification of CMG complex components suggests that each of these subunits is required for timely progression through S phase.

Proteasome activity promotes mitosis in *T. brucei*[69] and, consistent with this, proteasome components are particularly prominent amongst knockdowns that accumulate in $G_2M$; we illustrate the RIT-seq profiling data for this protein complex in Fig. 5c. Sixteen of 28 proteasome component knockdowns register >25% overrepresentation in the $G_2M$ pool. This output is consistent with the view that the *T. brucei* proteasome is responsible for degrading cell cycle regulators, such as poly-ubiquitinated cyclins, some of which are known to control cell cycle checkpoints in *T. brucei*. Candidate targets in *T. brucei* include: CIF1, AUK1[70], cyclin 6 (CYC6), degradation of which is required for mitosis[71]; cyclin-like CFB2, required for cytokinesis[72]; and cyclin 2 (CYC2) or cyclin 3 (CYC3), which have short half-lives and a candidate destruction box motif in the case of CYC3[73].

Kinetochore components[17] are also amongst knockdowns that accumulate in $G_2M$ and we illustrate the RIT-seq profiling data for this protein complex in Fig. 5d. Although knockdown of KKT2, a putative kinase, registered overrepresentation in the S phase pool, KKT1, KKT7 and KKT10/CLK1 knockdowns registered >25% overrepresentation in the $G_2M$ pool, suggesting that these particular kinetochore components, which all display temporal patterns of phosphorylation from S phase to $G_2M$[21], are required for progression through mitosis. Notably,

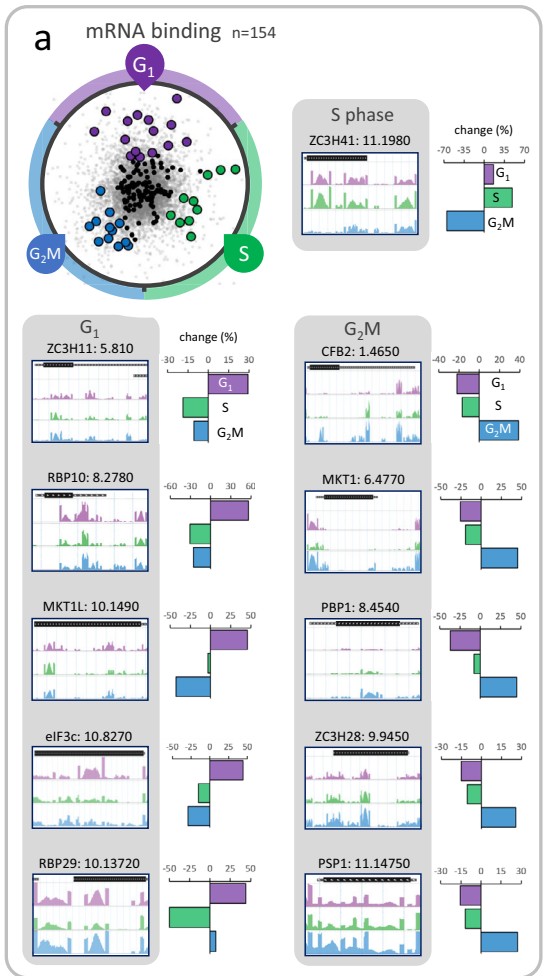
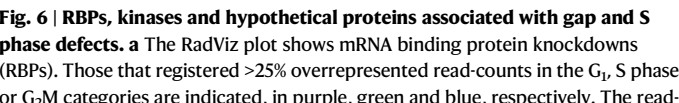

**Fig. 6 | RBPs, kinases and hypothetical proteins associated with gap and S phase defects. a** The RadViz plot shows mRNA binding protein knockdowns (RBPs). Those that registered >25% overrepresented read-counts in the $G_1$, S phase or $G_2M$ categories are indicated, in purple, green and blue, respectively. The read-mapping profiles and relative read-counts in the other panels show example hits. **b** As in **a** but for protein kinase knockdowns, with selected kinases indicated. **c** As in **a** but for hypothetical (conserved) protein knockdowns.

KKT10 is a kinase responsible for phosphorylation of KKT7, which is required for the metaphase to anaphase transition[74]; as well as for the phosphorylation of KKT1 and KKT2, in turn required for kinetochore assembly[75,76]. These findings are consistent with the view that kinetochore components control a non-canonical spindle checkpoint in trypanosomes[74].

### RBPs, kinases and hypothetical proteins associated with $G_1$, S phase and $G_2M$ defects

Widespread polycistronic transcription in trypanosomatids places great emphasis on post-transcriptional controls and, consistent with this, knockdowns overrepresented in the $G_1$, S phase and $G_2M$ pools revealed many putative mRNA binding proteins (RBPs) and kinases. Indeed, RBPs are significantly enriched amongst knockdowns that registered $G_1$, S phase or $G_2M$ cell cycle defects ($\chi^2$ test, $p = 7^{-5}$). We show the RIT-seq profiling data for eleven RBP knockdowns that register >25% overrepresentation in these pools (Fig. 6a). These include knockdowns for RBP10 and RBP29 enriched in $G_1$; RBP10, in particular, has been characterised in some detail and promotes the bloodstream form state[77]. ZC3H11[78], ZC3H41 and ZC3H28[79] knockdowns were enriched in $G_1$, S phase and $G_2M$, respectively, while knockdowns of CFB2, MKT1 or PBP1, all recently linked to variant surface glycoprotein expression control[80,81], were enriched in $G_2M$. Indeed, based on the outputs of the current screen, we prioritised these latter three RBPs for follow-up analysis in a separate study; all three were thereby validated

as $G_2M$ hits[80]. Thus, the RIT-seq cell cycle screen implicated a number of specific RBPs in post-transcriptional control of cell cycle progression through modulation of mRNA stability and/or translation.

We show data for protein kinases above, linked to enriched >4 C (Fig. 3d), S phase or $G_2M$ (Fig. 5d) phenotypes, and now show the RIT-seq profiling data for five additional protein kinase knockdowns that register >25% overrepresentation in the $G_1$, S phase or $G_2M$ pools (Fig. 6b). These include knockdowns for CRK7, linked to accumulation in $G_1$; MAPK5, linked to accumulation in S phase and polo-like kinase (PLK) and cdc2-related kinase 3 (CRK3), linked to accumulation in $G_2M$. PLK was previously shown to control cell morphology, furrow ingression and cytokinesis[82–84], while CRK3 was shown to play a role in $G_2M$ progression in bloodstream form *T. brucei*[43,85]. Overall correspondence was also excellent with a prior kinome-wide RNAi screen[43]. For example, eight among nine kinases linked to a mitosis defect in that screen also reported an ($21 \pm 12\%$) increase in the $G_2M$ pool in the current screen.

Finally, we analysed genes encoding proteins annotated as hypothetical (conserved). Despite excellent progress in genome annotation, 35% of non-redundant genes in *T. brucei* retain this annotation, amounting to >2500 genes. We show data for several hypothetical protein knockdowns above, linked to the enriched >4 C phenotype (Supplementary Fig. 4), and we here identify >300 additional hypothetical protein knockdowns that register >25% overrepresentation in the $G_1$, S phase or $G_2M$ pools. RIT-seq profiling data are shown for five examples in Fig. 6c and for several additional

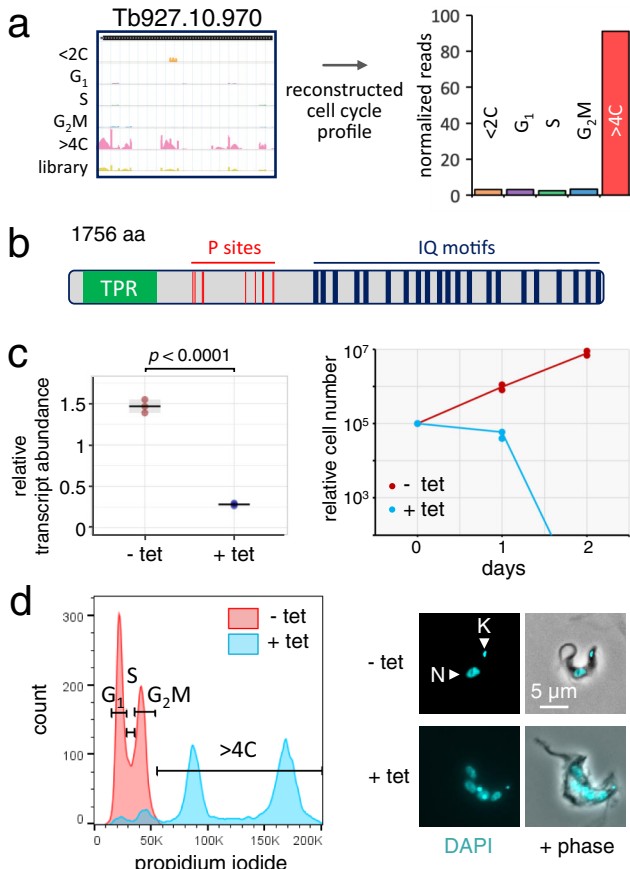

**Fig. 7 | Validation of endoreduplication following Tb927.10.970 knockdown. a** Mapped reads and reconstructed cell cycle profile following 10.970 knockdown. **b** Schematic map of 10.970. TPR tetratricopeptide repeat motif, P phosphorylation, IQ motifs, putative calmodulin-binding domains. **c** The plot on the left reveals efficient tetracycline (tet)-induced knockdown of 10.970 after 24 h. Mean values and 3 technical replicates are indicated, with the *p*-value derived using an unpaired *t*-test. The growth curve on the right reveals a severe growth defect following knockdown. $n = 2$ biologically independent clones and the line indicates mean values. Source data are provided as a Source Data file. **d** Flow cytometry analysis revealed endoreduplication following 24 h of 10.970 knockdown, based on propidium iodide staining. The gates indicate the various cell cycle stages, based on uninduced (- tet) cells. The panel on the right shows representative cells with DNA stained with DAPI (4',6-diamidino-2-phenylindole) before and after knockdown. N nucleus, K kinetoplast.

examples in Supplementary Fig. 5. Amongst other examples of knockdowns shown in Supplementary Fig. 5, are alternative oxidase[86], linked to $G_1$ enrichment; kinesins linked to $G_2M$ enrichment, including both chromosomal passenger complex kinesins (KIN-A and KIN-B)[26] and KIN-G; CYC6[25,87], centrin 3[88] and, finally, both components of the histone chaperone FACT (facilitates chromatin transcription) complex[89] Spt16 and Pob3, linked to $G_2M$ enrichment. Notably, the FACT complex has been linked to centromere function in human cells[90].

## Cell cycle regulated proteins linked to cell cycle progression defects

Factors required for cell cycle progression may themselves be cell cycle regulated. To identify some of these factors, we compared our current dataset with quantitative transcriptome[19], proteome[20] and phosphoproteome[21] cell cycle profiling data (Supplementary data 1). An initial survey of all 1,198 genes that registered a cell cycle defect here (see Fig. 2e) revealed significant enrichment of cell cycle regulated mRNAs (overlap = 114 of 484, $\chi^2$ $p = 3.2^{-4}$), and proteins displaying

cell cycle regulated phosphorylation (overlap = 112 of 547, $\chi^2$ $p = 0.025$). This, despite the fact that the transcriptome and (phospho) proteome datasets were derived from insect stage *T. brucei*, such that regulation may differ in the bloodstream *T. brucei* cells used for RIT-seq analysis here.

In terms of specific cell cycle regulated proteins[20] required for specific cell cycle progression steps, multiple glycolytic enzymes upregulated in $G_1$ were linked to accumulation in the $G_1$ pool following knockdown ($\chi^2$ $p = 7.9^{-11}$). In addition, proteins upregulated in $G_2$ and M were linked to accumulation in the $G_2M$ ($\chi^2$ $p = 1.8^{-8}$) or >4 C pools ($\chi^2$ $p = 8.9^{-9}$) following knockdown, including kinetochore and chromosomal passenger complex components, respectively. Some specific transcripts required for cell cycle progression may be upregulated prior to peak demand for the encoded protein, and we found evidence to support this view. For example, transcripts upregulated in late $G_1$ or in S phase were enriched amongst those knockdowns linked to accumulation in the $G_2M$ pool ($\chi^2$ $p = 3.3^{-3}$ and $p = 0.011$, respectively); both components of the FACT complex, upregulated in $G_1$, for example (see Supplementary Fig. 5). Similarly, S phase and $G_2M$ upregulated transcripts, including those encoding multiple flagellum-associated proteins, were enriched amongst knockdowns linked to accumulation in the >4 C pool ($\chi^2$ $p = 4.6^{-18}$ and $p = 2.4^{-5}$ respectively).

Some proteins displayed both cell cycle regulated expression and phosphorylation patterns that were consistent with their roles in cell cycle progression. These included putative RBPs of the DNA polymerase suppressor 1 (PSP1) family, which display mRNA upregulation in $G_1$, protein upregulation in S phase, cell cycle regulated phosphorylation and, following knockdown, accumulation in $G_2M$ (see Figs. 4c and 6a). The kinetochore components, KKT1 and KKT7, and also CRK3, all display mRNA upregulation in S phase, protein upregulation in $G_2$ and M, cell cycle regulated phosphorylation and, following knockdown, accumulation in $G_2M$ (see Figs. 5d and 6b); KKT10 and CYC6 report a similar profile (see Fig. 5d), except for the mRNA regulation component. The cytokinesis initiation factors, CIF1 and CIF2, display mRNA upregulation in S phase, protein upregulation in $G_2$ and M, cell cycle regulated phosphorylation and, following knockdown, accumulation in the endoreduplicated pool (see Fig. 3e). Finally, the chromosomal passenger complex components, CPC1 and AUK1, as well as furrow localized FRW1, report mRNA and protein upregulation in $G_2M$ and, following knockdown, accumulation in the endoreduplicated pool (see Fig. 3e). Thus, several regulators linked to specific cell cycle progression defects by RIT-seq profiling, are themselves cell cycle regulated.

## A putative nucleoredoxin controls kinetoplast segregation and mitosis

The current RIT-seq screen identified many novel candidate cell cycle regulators, two of which, both associated with significant loss-of-fitness in our prior RIT-seq screen[23], were investigated in more detail. First, Tb927.10.970 was associated with pronounced endoreduplication following knockdown (Fig. 7a). The predicted protein contains a tetratricopeptide repeat motif, a cluster of phosphorylation sites, one of which, $T^{706}$, has been reported to be cell cycle regulated, peaking in late $G_2M$[21], and a string of putative calmodulin-binding IQ domains (Fig. 7b). Tb927.10.970 was shown to localise to the paraflagellar rod in insect stage *T. brucei* (www.tryptag.org[91,92]). To validate Tb927.10.970 as a >4 C hit in bloodstream-form trypanosomes, we assembled a pair of independent inducible RNAi knockdown strains. Analysis of cell growth revealed a severe loss-of-fitness following knockdown, confirmed by qRT-PCR (Fig. 7c). Flow cytometry then confirmed endoreduplication, with prominent peaks detected representing 8 C and 16 C cells following knockdown (Fig. 7d, left-hand panel), while examination of these cells by microscopy revealed multiple nuclei, indicating endoreduplication with continued mitosis (Fig. 7d, right-hand panel).

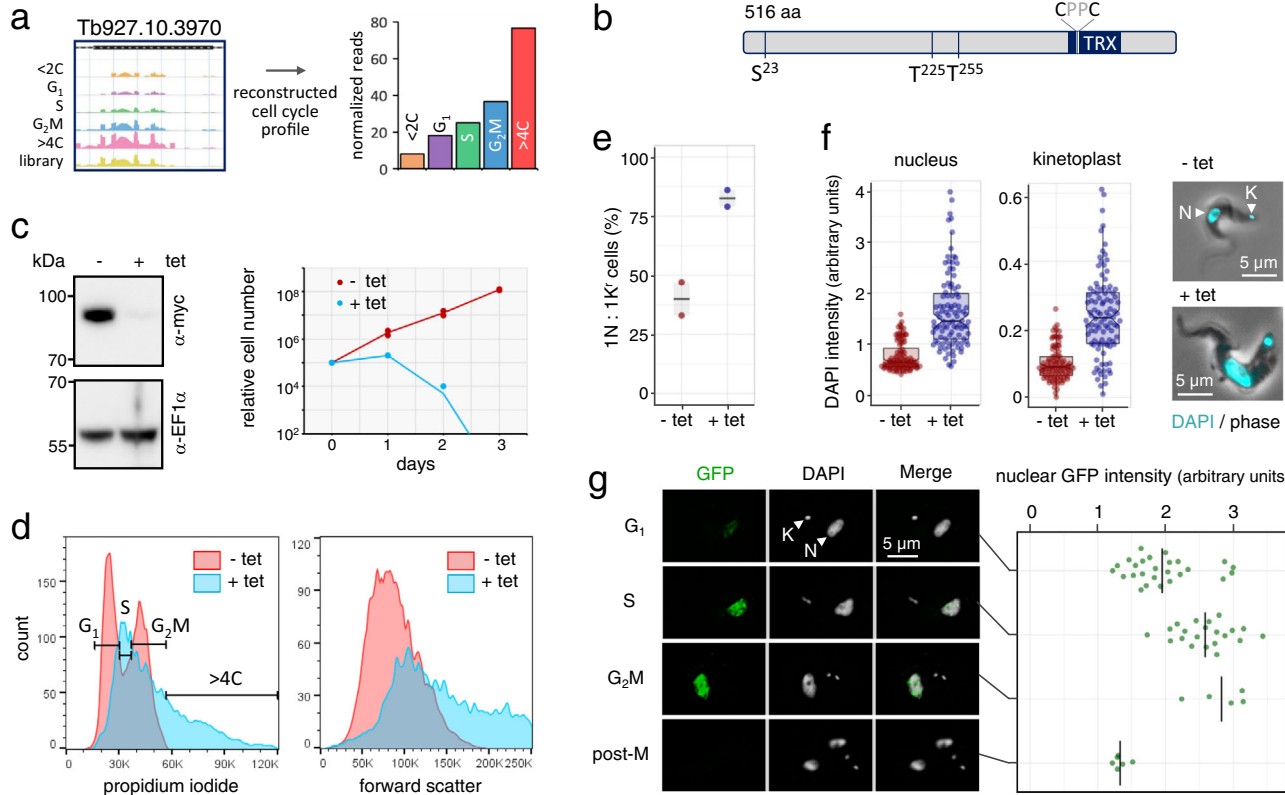

**Fig. 8 | A putative nucleoredoxin controls kinetoplast segregation and mitosis.** **a** Mapped reads and reconstructed cell cycle profile following 10.3970 knockdown. **b** Schematic map of the putative nucleoredoxin encoded by 10.3970. TRX thioredoxin-like domain. **c** The protein blots on the left reveal efficient tetracycline (tet)-induced knockdown of N-terminal myc-tagged 10.3970 for one representative clone after 24 h. EF1α serves as a loading control. The growth curve on the right reveals a severe growth defect following knockdown. $n = 2$ biologically independent clones and the line indicates mean values. **d** Flow cytometry analysis revealed increased DNA content following 24 h of 10.3970 knockdown, based on propidium iodide staining on the left, and increased cell size, based on the forward scatter profile on the right. The gates on the profile on the left indicate the various cell cycle stages, based on uninduced (- tet) cells. **e** Quantification of cells with a single nucleus and a single rounded kinetoplast (1 N: 1K$^r$) before and after 24 h of 10.3970 knockdown, based on DAPI-staining and microscopy. $n = 2$ biologically

independent clones and the lines indicate mean values; $n = 80$ cells in each case. **f** The boxplots show quantification of DAPI intensity of nuclei (left) and kinetoplasts (right) in cells with a single nucleus and a single rounded kinetoplast, before and after 24 h of 10.3970 knockdown; $n = 100$ in each case. The data are from two independent clones and are shown as jittered dots, the box indicates the IQR, whiskers show the range of values within 1.5*IQR, the horizontal lines indicate the median and the notches represent 95% confidence intervals. The inset on the right shows representative cells before and after knockdown. N, nucleus; K, kinetoplast. **g** The representative immunofluorescence microscopy images show N-terminal GFP-tagged 10.3970 (green) at different cell cycle stages, as defined by the nucleus (N) and kinetoplast (K) configuration observed by staining DNA with DAPI. Intensity of the $^{GFP}$10.3970 signal was quantified and is shown in the graph on the right. $n = 26$, 21, 5 and 6 cells for G₁, S phase, G₂M and post-M, respectively. The lines indicate the median. Source data for **c** and **e**–**g** are provided as a Source Data file.

We next turned our attention to Tb927.10.3970, annotated 'hypothetical protein, conserved', and associated with increased DNA content following knockdown (Fig. 8a). The predicted Tb927.10.3970 protein contains three cell cycle regulated phosphorylation sites[21] and a thioredoxin-like domain (Fig. 8b). This protein was shown to be cell cycle regulated based on proteomic analysis and to localise to the nucleus in insect stage *T. brucei*[20]. To explore the role of Tb927.10.3970 in bloodstream-form trypanosomes, we assembled a pair of independent inducible RNAi knockdown strains. Analysis of cell growth revealed a severe loss-of-fitness following knockdown, confirmed by monitoring the expression of epitope-tagged 10.3970 (Fig. 8c). Flow cytometry confirmed increased DNA content following knockdown, and also revealed increased cell size (Fig. 8d). Examination of these cells by microscopy allowed us to assess both the nuclei and kinetoplasts (mitochondrial genomes), revealing a major increase in the proportion of cells with a single nucleus and a single rounded kinetoplast following knockdown, an arrangement typically characteristic of G₁ cells (Fig. 8e). Quantitative analysis of these compartments revealed a pronounced increase in DNA content in both genomes following knockdown (Fig. 8f), indicating that both kinetoplast segregation and mitosis failed, while genome replication was able to proceed to varying degrees, generating enlarged

kinetoplasts and polyploid nuclei. Finally, we assessed the subcellular localisation of epitope-tagged 10.3970 during the bloodstream form cell cycle. Quantitative immunofluorescence microscopy revealed a pattern that was also observed in insect stage cells (www.tryptag.org[20,91,92]). 10.3970 displayed a nuclear localisation, which increased in intensity during the cell cycle, and then dropped precipitously in post-mitotic cells (Fig. 8g). We conclude that 10.3970 encodes a putative nucleoredoxin that is cell cycle regulated and required for both mitochondrial and nuclear genome segregation and cytokinesis.

## Discussion

Despite intense interest and study[13,15], many cell cycle regulators in trypanosomatids remain to be identified and much remains to be learned about cell cycle control and progression in these parasites. DNA staining followed by flow cytometry is a widely used approach for quantifying cellular DNA content and to analyse cell cycle distribution across otherwise asynchronous populations. Here, we combined genome scale loss-of-function genetic screening with DNA staining and flow cytometry in bloodstream form African trypanosomes and identify hundreds of genes required for progression through specific stages of the cell cycle.

Functional annotation of the trypanosomatid genomes will continue to benefit from novel high-throughput functional analyses, and RNAi-mediated knockdown has proven to be a powerful approach for *T. brucei*. RIT-seq profiling provides data for almost every gene and, using this approach, we previously described genome-scale loss-of-fitness data[23]. Amongst 3117 knockdowns that scored a significant loss-of-fitness in bloodstream-form cells in that screen (42% of all genes analysed) were genes encoding all 18 intraflagellar transport complex subunits ($\chi^2$ $p = 1^{-6}$), 12 of 13 dynein heavy-chains ($\chi^2$ $p = 4^{-4}$), all 8 TCP-1 chaperone components ($\chi^2$ $p = 1^{-3}$), 27 of 30 nucleoporins ($\chi^2$ $p = 2^{-7}$), all eleven glycolytic enzymes ($\chi^2$ $p = 2^{-4}$) and 30 of 31 proteasome subunits ($\chi^2$ $p = 2^{-9}$). This set also included 18 of 19 kinetochore proteins ($\chi^2$ $p = 6^{-6}$), only later identified as components of this essential complex[17]. With the current study, we now link many of these genes, and many more, to specific cell cycle defects following RNAi knockdown. A large number of flagellar protein knockdowns, in particular, yielded cells with excess DNA, indicating that DNA replication often continues following failure to complete cytokinesis, due to defects that occur during cytokinesis itself or earlier in the cell cycle in some cases. We identified pathways and protein complexes that impact cell cycle progression, such as glycolysis ($G_1/S$ transition) and the proteasome ($G_2/M$ transition). We also identify many mRNA binding proteins and protein kinases implicated in control of cell cycle progression. Notably, we link multiple known potential and promising drug targets to cell cycle progression defects, such as glycolytic enzymes[93], the proteasome[94], kinetochore kinases[75,95] and other kinases[96].

Prior cell cycle studies have often focused on trypanosome orthologues of known regulators from other eukaryotes. Since genome-scale profiling is unbiased, it presents the opportunity to uncover divergent as well as novel factors and regulators that impact cell cycle progression. Accordingly, we link many previously uncharacterised and hypothetical proteins of unknown function to specific cell cycle progression defects. Thus, we uncover mechanisms with an ancient origin in a common eukaryotic ancestor and others likely reflecting trypanosomatid-specific biology. We also compared our cell cycle profiling data with cell cycle regulated transcriptome and (phospho)proteome datasets.

The digital dataset provided in Supplementary data 1 facilitates further interrogation and further analysis of the genome-scale cell cycle RIT-seq data. We have also made the data available via an interactive, open access, online data visualization tool (https://tryp-cycle.pages.dev/), which allows data searching and browsing (see Supplementary Fig. 3). Comparison with existing and new datasets, including with high-throughput subcellular localisation data[92] www.tryptag.org, should facilitate future studies. Since high-throughput genetic screens typically yield a proportion of false positive 'hits'[22], we urge some caution, however, in particular where outputs may be predominantly generated by a single RIT-seq fragment. On the other hand, there are knockdowns in the current dataset that display potentially informative cell cycle pool enrichment yet fail to surpass the thresholds applied above. Considering both of these points, we hope that the digital dataset and the online tool will serve as valuable resources. Since other important trypanosomatid parasites, including other African trypanosomes, *Trypanosoma congolense, Trypanosoma vivax;* American trypanosomes, *Trypanosoma cruzi*; and *Leishmania* spp. share a high degree of conservation and synteny with *T. brucei* spp.[97] the current datasets can also assist and inform studies on these and other trypanosomatids.

Further illustrating the value of the current RIT-seq dataset, we analysed two novel hits from the screen in more detail and reveal a role for a putative nucleoredoxin in both kinetoplast and nuclear segregation. Indeed, although DNA replication continues, cells lacking this factor fail to undergo kinetoplast scission or separation[98], mitosis or cytokinesis. Mitochondrial and nuclear DNA replication and segregation are coordinated during the *T. brucei* cell cycle[13] to ensure inheritance of a single copy of each genome by each daughter cell, but the mechanism underlying coordination remains unknown. Identification of a thioredoxin-like protein required for both nuclear and kinetoplast DNA segregation suggests that redox signalling is involved in coordinating these processes. A comparable phenotype was observed following knockdown of both the mitochondrial and nuclear DNA-binding proteins, UMSBP1 (universal minicircle sequence binding protein) and UMSBP2, previously linked to replication and segregation of kinetoplast and nuclear DNA in insect stage *T. brucei*[99,100]. Notably, both DNA-binding and USMBP dimerization are redox-regulated[101]. Although not understood in detail, reactive cysteine thiols function as cell cycle associated redox sensors in mammalian cells[102]. Thus, we propose a role for the current putative nucleoredoxin in reducing thiols and potentially disrupting disulphide bonds in one or more key cell cycle regulators. This role may be conserved amongst trypanosomatids and indeed, redox signalling may play further roles in cell cycle control, since our screen linked two additional thioredoxin-like proteins to specific and distinct cell cycle defects; TRX2, a redox-regulated mitochondrial chaperone[103], was linked to accumulation in $G_1$, while the putative thioredoxin encoded by Tb927.9.12330 was linked to accumulation in $G_2M$ (Supplementary data 1). Further work will be required to explore how thiol-based redox switch[104] or sensing mechanisms choreograph cell cycle progression in the trypanosomatids.

In summary, we report RNAi induced cell cycle defects at a genomic scale and identify the *T. brucei* genes that underlie these defects. The outputs confirm known roles in cell cycle progression and provide functional annotation for many additional genes, including many with no prior functional assignment and many that are trypanosomatid-specific. As such, the data not only improve our understanding of cell cycle progression in these important and divergent pathogens but should also accelerate further discovery. Taken together, our findings facilitate genome annotation and provide comprehensive genetic evidence for the protein complexes, pathways and regulatory factors that facilitate and coordinate progression through the trypanosome cell cycle.

## Methods

### *T. brucei* growth and manipulation

The bloodstream form *T. brucei* RNAi library[22] was thawed in HMI-11 containing 1 µg.ml$^{-1}$ of blasticidin and 0.2 µg.ml$^{-1}$ of phleomycin and incubated at 37 °C in 5% $CO_2$. After approximately 48 h, six flasks, each containing $2 \times 10^7$ cells in 150 ml of HMI-11 as above, were prepared; 1 µg.ml$^{-1}$ of tetracycline was added to five of them, while one served as the non-induced control. The cells were grown under these conditions for 24 h. Bloodstream form *T. brucei* 2T1 cells[105] were grown as above in the presence of 1 µg.ml$^{-1}$ phleomycin and 1 µg.ml$^{-1}$ puromycin. These cells were transfected by electroporation as described[22] and selected with 2.5 µg.ml$^{-1}$ hygromycin (RNAi constructs) or 10 µg.ml$^{-1}$ blasticidin (myc or GFP tagging constructs). RNAi knockdown was induced with 1 µg.ml$^{-1}$ tetracycline.

### Flow cytometry

RNAi library samples were harvested by centrifugation for 10 min at 1000 *g*. Cells from each flask were then re-suspended in 25 ml of 1x PBS (pH 7.0) supplemented with 5 mM EDTA and 1 % FBS ("supplemented PBS"), centrifuged again for 10 min at 1000 *g*, and then re-suspended in 0.5 ml of supplemented PBS. To each cell suspension, 9.5 ml of 1 % formaldehyde in supplemented PBS was added dropwise, with regular vortexing. The cells were incubated for a further 10 min at room temperature and then washed twice in 10 ml of supplemented PBS using centrifugation as above. The cells were finally re-suspended at $2.5 \times 10^7$ per ml in supplemented PBS and were subsequently stored at 4 °C, in the dark. Fixed cells, $3 \times 10^8$ Tet-induced and $10^7$ uninduced, were centrifuged for 10 min at 1000 *g*, and re-suspended in 10 ml of

supplemented PBS containing 0.01% Triton X-100 (Sigma Aldrich). The cells were incubated for 30 min at room temperature, centrifuged for 10 min at 700 $g$ and washed once in 10 ml of supplemented PBS. The cells were then re-suspended in 4 ml of supplemented PBS containing 10 µg.ml$^{-1}$ of propidium iodide (Sigma Aldrich) and 100 µg.ml$^{-1}$ of RNaseA (Sigma Aldrich), and incubated for 45 min at 37 °C, in the dark; cells were subsequently kept on ice and in the dark. Immediately prior to sorting, the Tet-induced cells were filtered (Filcon Cup-type filter, 50 µm mesh, BD™ Medimachine) into 5 ml polystyrene round-bottom tubes (BD Falcon). Cells were sorted using the BD Influx™ (Becton Dickinson) cell sorter, with BD FACSort™ software, at the Flow Cytometry and Cell Sorting Facility in the School of Life Sciences, University of Dundee. The cells were sorted into pools of <2 C (~5 × 10$^5$ cells), 2 C (G$_1$, 1 × 10$^7$ cells), 2-4 C (S, 1 × 10$^7$ cells), 4 C (G$_2$M, 1 × 10$^7$ cells) and >4 C (~9 × 10$^5$ cells) based on their DNA content and collected into 50 ml Falcon tubes (BD Falcon); total sorting time was approx. 4 h. The 2 C, 2–4 C and 4 C sorted samples were then run on a FACS LSR Fortessa flow cytometry analyser for a post-sorting quality check. For the analysis of Tb927.10.970 or Tb927.10.3970 knockdowns, 1 × 10$^7$ cells were centrifuged for 10 min at 1000 $g$, washed in supplemented PBS. Cells were fixed for 10 min in 1% paraformaldehyde in supplemented PBS, washed and stored at 4 °C in supplemented PBS. Cells were pelleted for 10 min at 1000 $g$ and permeabilised at room temperature for 30 min in supplemented PBS plus 0.01% Triton X-100. Cells were washed once in supplemented PBS followed by centrifugation for 10 min at 700 $g$ and then stained in supplemented PBS with 10 µg.ml$^{-1}$ propidium iodide and 100 µg.ml$^{-1}$ RNAse A for 1 h at 37 °C. The samples were run on a BD FACSCanto (Becton Dickinson). FlowJo v10.7.1 was used for data analysis and visualisation.

### RNA interference target amplification

The five pools of Tet-induced, sorted cells as well as uninduced or induced, but unsorted cells, were lysed overnight at 56 °C in 50% (v/v) of Buffer AL (Qiagen) and 0.5 mg.ml$^{-1}$ of Proteinase K (Qiagen), to reverse formaldehyde crosslinking. Genomic DNA was then extracted using the DNeasy Blood and Tissue DNA extraction kit (Qiagen), according to the manufacturer's instructions, with the exception that each sample was eluted in 50 µl of Buffer AE. The whole sample (range = 140–840 ng) was used for PCR, in a 100 µl reaction, using OneTaq (NEB), and the Lib3F (CCTCGAGGGCCAGTGAG) and Lib3R (ATCAAGCTTGGCCTGTGAG) primers and with the following programme: 94 °C for 4 min, followed by 27 cycles of 94 °C for 30 sec, 55 °C for 30 sec and 68 °C for 2 min and 10 sec, and a final extension of 68 °C for 5 min. The PCR products were then purified using the Qiaquick PCR extraction kit (Qiagen), as per the manufacturer's instructions, and eluted in 30 µl of nuclease-free water (Ambion); two columns per sample.

### RIT-seq library preparation and sequencing

Purified PCR products were used for library preparation and sequencing at the Tayside Centre for Genomic Analysis at the University of Dundee. The PCR products were fragmented using a Covaris M220 sonicator (20% duty factor, 75 W peak/displayed power, 60 s duration – 3 × 20 sec with intermittent spin down step, 18–20 °C temperature; resulting in 250–300 bp enriched fragments), and the libraries were prepared using the Truseq Nano DNA Library Prep kit (Illumina). The samples were multiplexed, and sequenced on an Illumina NextSeq 500 platform, on a 150 cycle Output Cartridge v2, paired-end. Each library was run on 4 sequencing lanes. Base call, index deconvolution, trimming and QC were performed in BaseSpace using bcl2fastq2 Conversion Software v2.17.

### RIT-seq data mapping and analysis

The sequencing data analysis pipeline was adapted from[22]. The FASTQ files with forward and reverse paired end reads (4 technical replicates for each samples) were concatenated and aligned to the reference genome v46 of *T. brucei* clone TREU927 downloaded from TriTrypDB[29] using Bowtie2[106], with the 'very-sensitive-local' pre-set alignment option. The alignments were converted to BAM format, reference sorted and indexed with SAMtools[107]. The quality of alignments was evaluated with Qualimap 2[108] using the bamqc and rnaseq options. The Qualimap 2 output files were aggregated with MultiQC[109] and inspected. The alignments were deduplicated with the Picard tools package using the MarkDuplicates function (http://broadinstitute.github.io/picard/); to minimise the potential for overrepresentation of the shortest RIT-seq fragments. Alignments with properly paired reads were extracted with SAMtool view using the -f 2 option and parsed with a custom python script to extract the paired reads containing the barcode sequence (GTGAGGCCTCGCGA) in forward or reverse complement orientation. The genome coverage of the aligned reads was extracted from the bam files using deeptools in bedGraph format[110]. The–scaleFactor option was used to normalise each sample with respect to the library size. Firstly, the mean value of the library size was computed from all the samples. Secondly, the mean value was divided by the library size of each sample to obtain the scaling factors. The bedGraph read-mapping files were visualized with the svist4get python package[111]. Read counts for protein coding sequences and associated untranslated regions (where annotated) were determined from the bam files using featureCounts[112] and normalized to Transcripts Per Kilobase Million (TPM). Dimensionality reduction of the G$_1$, S and G$_2$M TPM values was performed with the radviz algoritm implemented in the pandas python package[113]. The bash script containing the analysis pipeline, a conda environment specification file for its execution, the python script to extract barcoded reads and a basic usage example are available in GitHub (https://github.com/mtinti/ritseq_cellcycle). Data were subsequently analysed using a GO-slim set and Gene Ontology tools available via tritrypdb.org and visualised using tools available at huygens.science.uva.nl/PlotsOfData.

### Plasmid construction for knockdown and tagging

For tetracycline-inducible knockdown of Tb927.10.970 or Tb927.10.3970 expression, gene fragments were amplified using either the 970RiF (GATCGGGCCCGGTACCCGCCACACTGAACAACCTT) and 970RiR (GATCTCTAGAGGATCCTCCCTTTGCCGCCTTACCAC) PCR primers or the 3970RiF (GATCGGGCCCGGTACCGCGTCGGAGATGTGATCCTT) and 3970RiR (GATCTCTAGAGGATCCACAACTCGCATACACGGAGG) PCR primers and cloned in two steps in pRPa$^{iSL}$ [105]. The resulting constructs were confirmed by sequencing and digested with AscI prior to transfection. Tb927.10.970 knockdown was assessed using Real-Time Quantitative Reverse Transcription PCR (qRT-PCR). RNA was extracted using the RNeasy Mini Kit with an on-column DNase digest step (RNase-free DNase, Qiagen). RNA (1 µg) was reverse-transcribed using a high capacity RNA to cDNA kit (Thermo Fisher Scientific) and the equivalent of 25 ng of RNA was used in each qPCR reaction, with the 970qRT_F (AGGAAGCGGAAGGAGAGGAT) and 970qRT_R (AGCGGAATTTATGCGTTCGC) primers. The reactions were performed in technical triplicates on a QuantStudio3 (Applied Biosystems) using the Luna® Universal qPCR Master Mix (NEB). TERT (Tb927.11.10190) was used as reference gene to calculate the 2^-deltaCt value for each sample. For N-terminal tagging of Tb927.10.3970, the targeting fragment was amplified using the 3970tagF (GATCTCTAGAGTAGGTGCTTCTTCCAAGC) and 3970tagR (GATCGGATCCCCGGAAGAGCACTAAGATCC) PCR primers and cloned into either 6 x myc or GFP pNAT$^{TAG}$x tagging constructs; versions with a blasticidin selectable marker were used[105]. Correct assembly was confirmed by sequencing and these constructs were linearised with SalI prior to transfection.

### Protein blotting

Pelleted cells were lysed in RIPA buffer with 8 × 5 s sonication cycles at 5 µm amplitude in a Soniprep 150 Utrasonic Disintegrator (MSE) with a

23 kHz generator and centrifuged at 4 °C for 15 min at maximum speed. Proteins in the supernatant were separated using SDS-PAGE and standard blotting procedures. Protein detection was achieved using mouse α-myc primary antibody clone 9B11 (1:5000) or mouse α-EF1α primary antibody (Millipore, 1:20,000) with goat α-mouse horseradish peroxidase coupled secondary antibody (Biorad, 1:2000). Blots were developed using the enhanced chemiluminescence kit (Amersham) as per the manufacturer's instructions.

## Microscopy and immunofluorescence microscopy

Cells were fixed in 1% paraformaldehyde and attached to 12-well 5 mm slides (Thermo Scientific) by drying overnight. Following rehydration in PBS for 5 min, cells were permeabilised in 0.5% Triton X-100 for 15 min and washed 3 x in PBS. Slides were blocked with 50% FBS in PBS for 15 min and washed in PBS twice. Slides were first incubated with rabbit α-GFP primary antibody (Abcam, 1:250) then α-rabbit Alexa 488 secondary antibody (1:2000) for 1 h each followed by 3 washes in PBS. Slides were finally mounted in Vectashield with DAPI (4′,6-diamidino-2-phenylindole) and sealed under a coverslip prior to imaging. For DAPI-only staining, slides were directly mounted in Vectashield with DAPI after rehydration. Cells were imaged at 63x magnification with oil immersion under a Zeiss Axiovert 200 M microscope with Zen Pro software (Zeiss). DAPI and GFP visualisation and quantification were performed in ImageJ (Fiji) v1.53q. For quantification of signal intensity, particles were defined automatically from a binary image of the DAPI channel then kinetoplast and nucleus particles were distinguished manually.

## Reporting summary

Further information on research design is available in the Nature Research Reporting Summary linked to this article.

## Data availability

The high-throughput sequencing data generated in this study have been deposited in the Short Read Archive (SRA) under accession code PRJNA641153. The mapped data can be visualised using an online tool at https://tryp-cycle.pages.dev/. Other data relating to individual genes can be found at tritrypdb.org. Source data are provided with this paper.

## Code availability

Code for the RIT-seq data analysis[114] (https://zenodo.org/record/7002689) and for the online visualisation tool[115] (https://zenodo.org/record/7002687) are available in GitHub.

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

## Acknowledgements

We thank R. Clark, A. Rennie and M. Lee of the Flow Cytometry and Cell Sorting Facility, which is supported by the Wellcome Trust (097418/Z/11/Z). We also thank L. Glover for advice on RNAi library manipulation, S. Hutchinson for advice on RIT-seq data analysis and J. Faria and G. Bravo Ruiz for fruitful discussions. The work was funded by Wellcome Trust Investigator Awards [100320/Z/12/Z and 217105/Z/19/Z to D.H.]. The funders had no role in study design, data collection and interpretation, or the decision to submit the work for publication.

## Author contributions

C.A.M., M.R., A.C. and D.H. designed the experiments. C.A.M. and M.R. performed the experiments. C.A.M., M.R., M.T. and D.H. analysed experimental data. M.T. performed data curation and visualisation. D.H. supervised the project and acquired funding. C.A.M., M.R., M.T. and D.H. wrote the manuscript.

## Competing interests

The authors declare no competing interests.
