## [Peer Review File · Nature Communications]

Reviewer comments , first round review -

Reviewer #1 (Remarks to the Author):

In "Genome-scale RNA interference profiling of *Trypanosoma brucei* cell cycle progression defects" by Marques et al., the authors perform a whole genome RNAi library screen in *T. brucei*. The original RIT-seq approach by the Horn lab was documented in Genome Research, and provided a very powerful tool for the *T. brucei* research community which has revolutionised some areas of research. Here, after induction of RNAi, cells are sorted according to cell cycle stage. Subsequently, it was determined which RNAi knock-downs led to cells accumulating at particular cell cycle stages. This approach is effective in picking up genes, which if knocked-down, don't cause significant death within the 24 hour time point used. One could argue that this might particularly be problematic for an experimental approach analysing the cell cycle, as a hard block in cell cycle progression could often lead to rapid death. However, the authors argue that these comprise a small subset of the population.

Knockdown of a significant proportion of the library (16.6% of 7,205 genes analysed) resulted in some form of perturbation in the cell cycle. This is presumably a consequence of secondary effects resulting in cells stalling. Many known proteins which influence the cell cycle (DOT1A, CIF1-4) were identified, arguing that the screen is robust within the large amount of noise caused by secondary effects. Glycolysis appears to be important for cell cycle progression through G1. This would argue that *T. brucei* senses nutrients and has metabolic control over the cell cycle, as has been documented in other organisms. Most of the known factors previously identified to play a role in cell cycle control are found here too. It is however less clear how many of the hits are false positives, as only two hits from the screen including a putative nucleoredoxin are validated in more detail.

The screen hasn't yet given us novel insights into cell cycle progression in *T. brucei*, with for example the identification of novel cell cycle checkpoints. In addition, the very large number of genes identified (1,197) could limit its use to specifically identify cell cycle regulators. However, the manuscript does provide a very extensive description of what should be a very useful resource for the *T. brucei* research community, which would need to be further investigated in more detail.

One of the few specific cell cycle arrests previously described in bloodstream form *T. brucei* is the very specific precytokinesis cell cycle arrest which occurs when Variant Surface Glycoprotein (VSG) synthesis is blocked, arguing that a checkpoint is triggered. The resulting stalled cells remain viable for several days. It was surprising that VSG or genes involved in VSG synthesis did not appear to be identified in the G2/M fraction in this screen. Is VSG221 present in the RNAi library used? Possibly not, as telomeric fragments are typically underrepresented in standard restriction enzyme based libraries. It is also possible that knock-down of genes involved in VSG synthesis is too lethal (within the 0.6% of cells discussed) to be picked up. I think that this point needs to be discussed.

Further:

-Some of the GO terms were not easily interpretable. For instance in Fig. 3A and Fig. 4A in the lower G2/M panel, the GO terms "mitotic cell cycle", "cell cycle" and "regulation of cell cycle" are used. Is there a way of combining or omitting vague/ redundant GO terms? Or alternatively explain why they are being labelled like this?

-Line 148 and 152 refer to 'high-speed' cell sorting. What is there about this type of cell sorting that makes it high-speed rather than regular speed? This was not clear from the text.

-Sup. Figure S3. The figure of the RadViz plot might be more clear with a dot indicating the centre of the circle, and concentric circles providing a scale for relative read counts.

-It was unclear how the online visualisation tool was annotated. Why does beta-tubulin (Tb927.1.2330) have a blue dot labelled G2/M when the mapped reads (pink) are primarily in GG2M?

Reviewer #2 (Remarks to the Author):

The trypanosome cell cycle has not been subjected to global analysis by forward mutational genetic screens since such approaches are impossible in these organisms. This type of global post-genomic approach is therefore very welcome and will be an excellent and valuable on-going resource – particularly when used in connection with other similar global analyses of proteins, genes, expression patterns.

The work is well done and well discussed by considering the previous multiple individual or small group gene studies. These “validations” are useful in bolstering confidence in the results and should not be seen as merely repeating available hits. Whilst there is no one major discovery in the data it is nevertheless hugely important and enabling. The authors do give a nice example of how such data will be enabling of much future work in many laboratories.

General Points:

Given the transfection of the library can the authors comment on the statistics of transfection. Are these clones likely to only contain one RNAi sequence per cell? Is there any chance of combinatorial effects?

Are there any phenotypes that could be interpreted as cells entering G zero - ie. differentiation induction?

Do the authors believe they are dealing with a 'block' at a stage in the cell cycle or merely extending the length of a particular stage? Can they distinguish between these by analysis using further flow cytometry in some cases?

Supplementary Figure 3. In the first line of the legend the URL for the full dataset is given. Looking at this site I can see the circle with dots but I found it difficult [I admit I am speaking here for the non - experts!] to get a firm idea of the confidence levels of the conclusions within the dataset as a whole - and hence the rationale for dot coloring.

For instance - if one looks at two dots with a large radial value - Tb 927.10. 9660 - the furthest out orange (<2C LG1) dot the transcript profile does indeed suggest a high LGI read count category in the window that comes up on mousing over. However, the green (S) dot with the largest radial position - Tb 927.10.9350 - has an almost identical profile of S and G1 read counts, plus in fact a good deal of G2M signal. So, why S?

How and why are such positions and coloring achieved?

What is the threshold for each?

How are is the inclusion in one category defined?

Would it be possible to add some explanation to this central dataset and more importantly some confidence form of banding to the data (a shaded set of circular confidence thresholds, as background? Contour lines?).

Specific Points:

Line 195/196. This is ambiguous. Cytoplasmic dynein does move along microtubules but axonemal versions are tethered and 'slide' along adjacent microtubules only to some extent producing bending. A better, more precise description is required.

What are GG2M and LG1 categories in the figures? I couldn't see the definition (sorry if I missed it but also could not find it by searching the PDF of this paper? Hence, I found it difficult to assess some of this data.

Line 377/378. These findings relating to metabolism and G1 are very much what one would expect in terms of cell cycle and nutrient sensing - but it is still an important aspect of this work that this is established for trypanosomes. It again raises the earlier question as to whether such metabolic stringency leads to a block (equivalent to yeast start) in the cycle or merely an extension of that G1 period. Is there a way that, for some selected genes in this category, an individual cell cycle

analysis of some of these genes might shed light on this matter - would additional simple flow cytometry of some mutants be useful? Again, I may have missed this type of analysis in the data.

Two examples are taken for global analysis.

Tb 927. 10. 970 - endo reduplication following knockdown. The authors say this localizes to the PFR and has phosphorylation cell cycle regulation with references. The global analysis in TrypTag <http://www.tryptag.org> already shows it to be a PFR protein and shows the presence of the protein at all stages of the cell cycle. Cell cycle regulation of the protein itself (not debating the phosphorylation evidence) is likely only to be in the aspect that a new flagellum is formed each cycle. Hence, a peak in G2M would be indicative of 2 flagella per cell versus 1 in the G1 phase?

Tb927. 10.3970 – shows enlarged kinetoplasts and nuclear location with a rapid drop in post mitotic cells. Again, in the global TrypTag dataset <http://www.tryptag.org> this protein has been clearly defined as located in the nucleus and showing a characteristic regulation with rapid post mitotic disappearance in the cell cycle analysis on that site.

I mention these issues not to argue that this detracts from the insights of this paper but it seems that perhaps this published data from TrypTag, another global analysis, should be referenced for completeness and to signify the importance of these modern global analyses conducted by various groups. I would hope the many user of this dataset will also reference accordingly.

Reviewer #3 (Remarks to the Author):

In this work, Marque and Ridgway et al have developed a novel implementation of the RIT-seq whole genome RNAi approach to uncover *T. brucei* proteins that play a role during the parasite cell cycle. By sorting cells based off of their DNA content, the authors were able to identify where in the cell cycle certain RIT-seq RNAi targets were enriched, which suggests what cell cycle stage transition they were necessary for. Using this approach, the authors were able to identify 1197 genes that registered a cell cycle defect, including many proteins with known roles and a whole host of new candidate cell-cycle regulators, including a putative nucleoredoxin that is essential for kinetoplast segregation and mitosis. They also discover pathway-level involvement of glycolysis in the G1 to M transition. Overall, the work provides a valuable dataset with many novel gene candidates to interrogate for function in cell cycle progression. The follow-up work on 10.970 and 10.3970 does not provide much mechanistic detail but does highlight the possibilities of the gene candidates presented.

My main criticisms focus on the presentation of the data and the specifics of how the cells were sorted

For the cell sorting:

1. Was the induced population of the cells checked for apoptotic or necrotic states prior to fixation for sorting? Highly toxic RNAi segments could have triggered these events, which may look like cell cycle blocks to the flow cytometer.

2. While zoids may be unusual in BSFs, it seems that flow cytometry/sorting would be an optimal approach for capturing rare events. Was there any evidence of very small/very dim cells in the flow data?

For presentation:

1. Considering the timing and geometric constraints on cell division in *T. brucei*, it is common for a defect in an upstream cell division event to have an indirect effect on cytokinesis. A good example is PLK, whose activity is only essential during the early stages of the cell cycle but is dispensable during cytokinesis. PLK inhibition or RNAi causes defects in new FAZ formation, which blocks the recruitment of the CIFs and other cytokinetic proteins as a secondary, or even tertiary effect.

Similarly, although some of the RIT-seq hits appear to have blocked cytokinesis, their absence may have perturbed a much earlier event. I think this is an important point that should be mentioned- that cytokinesis defects can function as a "catch-basin" for many other upstream defects, especially when your only readout is DNA content.

2. It seems strange to use the exocyst as negative control for a cell division screen, considering that it is heavily involved in the later stages of cytokinesis in mammals and yeasts. I realize the authors have looked at exocyst components in *T. brucei* in different work, but I don't know if has been shown conclusively that the *T. brucei* exocyst isn't involved in cell division.

3. It should be noted that BSFs appear to be able to fail to complete cytokinesis, restart the cell cycle, and complete the process from an 8C state (Wheeler Mol Micro 2013). This might be worth mentioning- perhaps some of the hits in the screen move cells towards this cell division mode rather than the more conventional one.

More minor concerns:

Figure 1A: Could the authors include a schematic branching off the >4C category representing the phenotype described in Figure 8? A cell without multiple furrows but an enlarged nucleus would make the point well.

Line 108 : "required G1S progression" should read "required [for] G1S progression"

Line 155: could you sort zoids, or did you observe them? Were there many cells outside of the largest or smallest gates?

Line 258: Exocyst is essential during the latest stages of cell division in other organisms.

Line 308: It might be worth mentioning specifically that flagellar beat has been implicated in the last stages of cytokinesis- the trypanin work by Kent Hill is cited but I think you could make a stronger case for why the motile flagellum is important.

Line 321: It would be good to have more information about the <2C cells. I realize this is a small set of genes- but is there any unifying theme in terms of function or localization? DOT1 does not appear to have the orange shading expected in the supplemental data, unless I misunderstand how the data is supposed to be presented.

20th July 2022

Genome-scale RNA interference profiling of *Trypanosoma brucei* cell cycle progression defects
Nature Communications

Point-by-point response to the reviewers' comments.

Reviewer 1:

1.1: In “Genome-scale RNA interference profiling of *Trypanosoma brucei* cell cycle progression defects” by Marques et al., the authors perform a whole genome RNAi library screen in *T. brucei*. The original RIT-seq approach by the Horn lab was documented in Genome Research, and provided a very powerful tool for the *T. brucei* research community which has revolutionised some areas of research. Here, after induction of RNAi, cells are sorted according to cell cycle stage. Subsequently, it was determined which RNAi knock-downs led to cells accumulating at particular cell cycle stages. This approach is effective in picking up genes, which if knocked-down, don't cause significant death within the 24 hour time point used. One could argue that this might particularly be problematic for an experimental approach analysing the cell cycle, as a hard block in cell cycle progression could often lead to rapid death. However, the authors argue that these comprise a small subset of the population.

R1.1: Indeed, our experimental design effectively mitigated against this potential problem - “reads for only 0.6% of genes dropped by >3-fold following 24 h of knockdown in the ... samples analysed here” (see 4th paragraph of 1st Results section). As a result, the approach yielded potentially informative data for the vast majority of *T. brucei* genes. It's also worth noting here that for those genes where reads do drop >3-fold, we can still obtain informative data, as was the case for four of eight α/β subunits of the proteasome that registered a G₂M defect, for example.

1.2: Knockdown of a significant proportion of the library (16.6% of 7,205 genes analysed) resulted in some form of perturbation in the cell cycle. This is presumably a consequence of secondary effects resulting in cells stalling. Many known proteins which influence the cell cycle (DOT1A, CIF1-4) were identified, arguing that the screen is robust within the large amount of noise caused by secondary effects. Glycolysis appears to be important for cell cycle progression through G1. This would argue that *T. brucei* senses nutrients and has metabolic control over the cell cycle, as has been documented in other organisms. Most of the known factors previously identified to play a role in cell cycle control are found here too. It is however less clear how many of the hits are false positives, as only two hits from the screen including a putative nucleoredoxin are validated in more detail.

R1.2: We agree that some results from our screen report ‘secondary’ effects but don't feel that these effects should necessarily be simply considered a source of ‘noise’. As an example, proteasome loss-of-function triggers accumulation at G₂M “consistent with the view that the *T. brucei* proteasome is responsible for degrading cell cycle regulators” (see 3rd paragraph in the ‘Pathways ...’ Results section). In terms of false positives, we now cite a manuscript (Glover *et al*, *Nature Protocols*. 2016) where we discussed ‘Potential sources of false negative and false-positive results’ in RIT-seq screens in more detail, immediately after our current text stating “high-throughput genetic screens typically yield a proportion of false positive ‘hits’” (see 4th paragraph in the Discussion).

1.3: The screen hasn't yet given us novel insights into cell cycle progression in *T. brucei*, with for example the identification of novel cell cycle checkpoints. In addition, the very large number of genes identified (1,197) could limit its use to specifically identify cell cycle regulators. However, the manuscript does provide a very extensive description of what should be a very useful resource for the *T. brucei* research community, which would need to be further investigated in more detail.

R1.3: We agree that many hypotheses that emerge from the current screen will require follow-up analysis. As detailed in the 4th paragraph of our Discussion, “Comparison with existing and new datasets, including with high-throughput subcellular localisation data www.tryptag.org, should facilitate future studies” and also “we hope that the digital dataset and the online tool will serve as valuable resources”. We’ve now edited our abstract to highlight areas where we feel that the screen has yielded novel insights, including “metabolic control of the G₁-S transition, surface antigen regulatory mRNA-binding proteins required for cytokinesis, and a putative nucleoredoxin required for both mitochondrial genome segregation and for mitosis”.

1.4: One of the few specific cell cycle arrests previously described in bloodstream form *T. brucei* is the very specific precytokinesis cell cycle arrest which occurs when Variant Surface Glycoprotein (VSG) synthesis is blocked, arguing that a checkpoint is triggered. The resulting stalled cells remain viable for several days. It was surprising that VSG or genes involved in VSG synthesis did not appear to be identified in the G₂/M fraction in this screen. Is VSG221 present in the RNAi library used? Possibly not, as telomeric fragments are typically underrepresented in standard restriction enzyme based libraries. It is also possible that knock-down of genes involved in VSG synthesis is too lethal (within the 0.6% of cells discussed) to be picked up. I think that this point needs to be discussed.

R1.4: Genes involved in VSG synthesis were indeed identified in the G₂/M fraction in this screen: “CFB2, MKT1 and PBP1, all recently linked to variant surface glycoprotein expression control^{80,81}, were enriched in G₂/M” (see 1st paragraph in the ‘RBPs ...’ Results section). We’ve now followed this text with more detail to emphasise this point: “Indeed, we prioritised these latter three RBPs for follow-up analysis in a separate study⁸⁰ based on the outputs of the current screen”. VSG221 is not present in the RNAi plasmid library as this was assembled using genomic DNA from the ‘927’ reference strain (see²² and Morris *et al.*, 2002; PMID: 12198145), which has a divergent set of VSGs.

Further:

1.5: Some of the GO terms were not easily interpretable. For instance in Fig. 3A and Fig. 4A in the lower G₂/M panel, the GO terms “mitotic cell cycle”, “cell cycle” and “regulation of cell cycle” are used. Is there a way of combining or omitting vague/ redundant GO terms? Or alternatively explain why they are being labelled like this?

R1.5: Indeed, there is substantial redundancy amongst the cohorts of genes underpinning these three terms. We have now removed the terms “mitotic cell cycle” and “regulation of cell cycle” from the plots in Fig. 3A and Fig. 4A.

1.6: Line 148 and 152 refer to ‘high-speed’ cell sorting. What is there about this type of cell sorting that makes it high-speed rather than regular speed? This was not clear from the text.

R1.6: Thank-you for raising this point. We used the BD Influx sorter for high-throughput and “high-speed sorting”, which was enabled by the design of the nozzle assembly (<https://www.bd.com/resource.aspx?IDX=17866>). However, we now see that the speed achieved for our experiment is not now considered to be “high-speed”, reflecting technological advances in flow cytometry in recent years. We have therefore removed those references to “high-speed” sorting referred to above.

1.7: Sup. Figure S3. The figure of the RadViz plot might be more clear with a dot indicating the centre of the circle, and concentric circles providing a scale for relative read counts.

R1.7: We have now added a marker indicating the centre of the circle, and concentric circles to the RadViz plot, as suggested (see Fig. S3 and <https://tryp-cycle.pages.dev/>).

1.8: It was unclear how the online visualisation tool was annotated. Why does beta-tubulin (Tb927.1.2330) have a blue dot labelled G₂/M when the mapped reads (pink) are primarily in GG₂/M?

R1.8: Tb927.1.2330 scores as a hit in both the G₂/M and >4C (aka ‘GG₂/M’, but see R2.8 below) categories (see Additional file 1). Indeed, 165 genes among 1,198 (14%) score as hits

in >1 category (see Fig. 2e), but we are unable to assign >1 colour to each hit in the online visualisation. We've now modified the online visualization tool (see R2.4 and 2.5 below) and have added some explanatory text, also stating "All genes have been assigned to a class where they register an elevated read-count". In relation to tubulin specifically, Tb927.1.2330 does in fact now register as a pink dot in the modified online visualisation and we've also now modified the right-hand panels in Fig. 2A to show read-count averages with error-bars for the α/β tubulin genes.

Reviewer 2:

2.1: The trypanosome cell cycle has not been subjected to global analysis by forward mutational genetic screens since such approaches are impossible in these organisms. This type of global post-genomic approach is therefore very welcome and will be an excellent and valuable on-going resource – particularly when used in connection with other similar global analyses of proteins, genes, expression patterns.

The work is well done and well discussed by considering the previous multiple individual or small group gene studies. These "validations" are useful in bolstering confidence in the results and should not be seen as merely repeating available hits. Whilst there is no one major discovery in the data it is nevertheless hugely important and enabling. The authors do give a nice example of how such data will be enabling of much future work in many laboratories.

R2.1: We thank the reviewer for their positive comments on the importance of our results and data and the likely impact as an enabling resource.

General Points:

2.2: Given the transfection of the library can the authors comment on the statistics of transfection. Are these clones likely to only contain one RNAi sequence per cell? Is there any chance of combinatorial effects?

R2.2: The clones are likely to contain only one RNAi sequence per cell. We've added a comment to clarify and have also cited the relevant studies where this issue is discussed in more detail; "This is achieved by targeting each cassette to a specific, single chromosomal locus that supports robust and reproducible inducible expression^{22,27}" (see 2nd paragraph in the Results section).

2.3: Are there any phenotypes that could be interpreted as cells entering G zero - ie. differentiation induction?

R2.3: We discuss this possibility in relation to glycolysis: "The results are also consistent with the observation that *T. brucei* accumulate in G₁ or G₀ under growth-limiting conditions⁶³ or during differentiation to the non-dividing stumpy form⁶⁴, possibly reflecting a role for glucose sensing in differentiation⁶⁵" (see 1st paragraph in the 'Pathways ...' Results section). Knockdown of RBP10 (Tb927.8.2780), which "promotes the bloodstream form state" (see 1st paragraph in the 'RBPs ...' Results section), also leads to accumulation in the G₁ pool (see Fig. 6A), which could also potentially reflect a G₀-like state. G₀ is indistinguishable from G₁ in our screen, however, such that we feel that further comment on this point in the manuscript could be too speculative.

2.4: Do the authors believe they are dealing with a 'block' at a stage in the cell cycle or merely extending the length of a particular stage? Can they distinguish between these by analysis using further flow cytometry in some cases?

R2.4: We believe we are dealing with a continuum between these two scenarios. We suspect that depletion is often tolerated up to a point, before which a particular stage is extended, and after which viability is typically lost, in a protein / pathway dependent manner. As one example, where we believe RIT-seq profiles provide some insight, our data "suggested that a reduced rate of DNA replication can be tolerated, albeit extending S phase" (see 4th paragraph in the Results section). Given the likely kinetics detailed above, we believe that distinguishing between these scenarios using further flow cytometry with individual knockdowns would

present several challenges. Indeed, any RNAi-induced differences observed may be substantially due to variable protein knockdown kinetics and penetrance, correlated with protein turnover rates to some degree. As such, we suggest that these issues would be better addressed, as appropriate, as part of follow-up studies.

2.5: Supplementary Figure 3. In the first line of the legend the URL for the full dataset is given. Looking at this site I can see the circle with dots but I found it difficult [I admit I am speaking here for the non - experts!] to get a firm idea of the confidence levels of the conclusions within the dataset as a whole - and hence the rationale for dot coloring. For instance - if one looks at two dots with a large radial value - Tb 927.10. 9660 - the furthest out orange (<2C LG1) dot the transcript profile does indeed suggest a high LGI read count category in the window that comes up on mousing over. However, the green (S) dot with the largest radial position - Tb 927.10.9350 - has an almost identical profile of S and GI read counts, plus in fact a good deal of G2M signal. So, why S?

R2.5: We have now modified the online visualization tool and Fig. S3 and added some explanatory text, which we hope helps to clarify (also see R1.7-1.8 above). Based on our analysis, Tb927.10.9660 registers as a G₁ hit (albeit also registering a strong <2C signal - see Additional file 1) and now also registers as a G₁ hit in our modified online visualization. Tb927.10.9350 does indeed register as an S phase hit (in both Additional file 1 and the online tool). To facilitate interpretation, we've now applied normalisation to these read-mapping plots (see 'RIT-seq data mapping and analysis' section in Methods) in the online tool and in all such panels in the main and Supplementary Figures. Confidence in individual outputs can be further assessed by scoring the number of RNAi target fragments reporting a particular phenotype (alongside the digital data in Additional file 1). To address this point we've added the following text to the online tool: "This app allows easy navigation of the cell cycle RIT-seq dataset (also see Marques et al., 2022 and Additional file 1 therein). The Radviz visualization on the left pulls the genes (data-points) closer to the 'time-point' where they show maximum accumulation. The read-coverage plots appear as tooltips after hovering over data-points in the Radviz plot or over the gene table. These coverage plots can be used to assess the number of different RNAi target fragments (that support a given phenotype) for each gene (also considering the untranslated regions), by counting the number of paired stacks of barcoded reads flanking each fragment. For example, the 300 bp 10.10030 CDS appears to be targeted by a single fragment, while the average-length 1.6 kbp 9.12810 CDS appears to be targeted by seven or more fragments. All genes have been assigned to a class where they register an elevated read-count".

2.6: How and why are such positions and coloring achieved?

What is the threshold for each?

How are is the inclusion in one category defined?

Would it be possible to add some explanation to this central dataset and more importantly some confidence form of banding to the data (a shaded set of circular confidence thresholds, as background? Contour lines?).

R2.6: These issues are addressed in modified Fig. S3 and in the modified online visualization tool with new associated text (also see R1.7-1.8 and R2.5 above).

Specific Points:

2.7: Line 195/196. This is ambiguous. Cytoplasmic dynein does move along microtubules but axonemal versions are tethered and 'slide' along adjacent microtubules only to some extent producing bending. A better, more precise description is required.

R2.7: Edited as suggested and now including "or drive microtubule sliding".

2.8: What are GG2M and LG1 categories in the figures? I couldn't see the definition (sorry if I missed it but also could not find it by searching the PDF of this paper? Hence, I found it difficult to assess some of this data.

R2.8: We apologise for leaving these earlier 'Less than G1' and 'Greater than G2M' labels in the coverage plots and have now removed all instances of 'LG1' and 'GG2M'. We now use '<2C' and '>4C' throughout.

2.9: Line 377/378. These findings relating to metabolism and G1 are very much what one would expect in terms of cell cycle and nutrient sensing - but it is still an important aspect of this work that this is established for trypanosomes. It again raises the earlier question as to whether such metabolic stringency leads to a block (equivalent to yeast start) in the cycle or merely an extension of that G1 period. Is there a way that, for some selected genes in this category, an individual cell cycle analysis of some of these genes might shed light on this matter - would additional simple flow cytometry of some mutants be useful? Again, I may have missed this type of analysis in the data.

R2.9: We don't believe that simple flow cytometry analysis would shed much light on this matter. Rather, a more detailed study would be required and would likely come with some challenges, some of which are outlined above (see R2.4).

2.10: Two examples are taken for global analysis.

Tb 927. 10. 970 - endo reduplication following knockdown. The authors say this localizes to the PFR and has phosphorylation cell cycle regulation with references. The global analysis in TrypTag <http://www.tryptag.org> already shows it to be a PFR protein and shows the presence of the protein at all stages of the cell cycle. Cell cycle regulation of the protein itself (not debating the phosphorylation evidence) is likely only to be in the aspect that a new flagellum is formed each cycle. Hence, a peak in G2M would be indicative of 2 flagella per cell versus 1 in the G1 phase?

Tb927. 10.3970 – shows enlarged kinetoplasts and nuclear location with a rapid drop in post mitotic cells. Again, in the global TrypTag dataset <http://www.tryptag.org> this protein has been clearly defined as located in the nucleus and showing a characteristic regulation with rapid post mitotic disappearance in the cell cycle analysis on that site.

I mention these issues not to argue that this detracts from the insights of this paper but it seems that perhaps this published data from TrypTag, another global analysis, should be referenced for completeness and to signify the importance of these modern global analyses conducted by various groups. I would hope the many user of this dataset will also reference accordingly.

R2.10: We have now cited the TrypTag database in relation to Tb927.10.970 localisation and apologise for erroneously citing reference ²⁰ for this protein previously (see final Results section). Reference ²⁰ (Crozier *et al.*, 2018) is still cited in relation to Tb927.10.3970 as that manuscript shows TrypTag data and proteomic analysis for this protein. We also agree with the reviewer in relation to the importance of the TrypTag dataset and now cite www.tryptag.org, Halliday *et al.*, 2019, and Dean *et al.*, 2017 in relation to each of these two genes; www.tryptag.org and Halliday *et al* were previously cited in our Discussion.

Reviewer 3:

In this work, Marque and Ridgway et al have developed a novel implementation of the RIT-seq whole genome RNAi approach to uncover *T. brucei* proteins that play a role during the parasite cell cycle. By sorting cells based off of their DNA content, the authors were able to identify where in the cell cycle certain RIT-seq RNAi targets were enriched, which suggests what cell cycle stage transition they were necessary for. Using this approach, the authors were able to identify 1197 genes that registered a cell cycle defect, including many proteins with known roles and a whole host of new candidate cell-cycle regulators, including a putative nucleoredoxin that is essential for kinetoplast segregation and mitosis. They also discover pathway-level involvement of glycolysis in the G1 to M transition. Overall, the work provides a valuable dataset with many novel gene candidates to interrogate for function in cell cycle

progression. The follow-up work on 10.970 and 10.3970 does not provide much mechanistic detail but does highlight the possibilities of the gene candidates presented.

My main criticisms focus on the presentation of the data and the specifics of how the cells were sorted. For the cell sorting:

3.1: Was the induced population of the cells checked for apoptotic or necrotic states prior to fixation for sorting? Highly toxic RNAi segments could have triggered these events, which may look like cell cycle blocks to the flow cytometer.

R3.1: The induced population was not checked for apoptotic or necrotic states prior to fixation. Such states may indeed look like cell cycle blocks to the flow cytometer, but nonetheless, we consider any such knockdowns that accumulate at specific cell cycle phases to be equally valid read-outs from the current screen. Since programmed cell death pathways remain controversial in parasitic protozoa, including *T. brucei* (see these two reviews: Proto *et al.*, 2013 - PMID: 23202528, Figueiredo and Menna-Barreto, 2019 - PMID: 30700697), we have not commented further on this topic in the manuscript.

3.2: While zoids may be unusual in BSFs, it seems that flow cytometry/sorting would be an optimal approach for capturing rare events. Was there any evidence of very small/very dim cells in the flow data?

R3.2: The relevant flow cytometry profiles are shown in Fig. S1A and zoids are discussed in the 'Defects producing sub-diploid cells' Results section. Ultimately, "any zoids present in the <2C pool will not have been detected using RIT-seq, since detection relies upon the presence of a nuclear RNAi target fragment". We've now also added "anuclear zoids are not shown as they are undetectable by RIT-seq" to the Fig. 1b legend.

For presentation:

3.3: Considering the timing and geometric constraints on cell division in *T. brucei*, it is common for a defect in an upstream cell division event to have an indirect effect on cytokinesis. A good example is PLK, whose activity is only essential during the early stages of the cell cycle but is dispensable during cytokinesis. PLK inhibition or RNAi causes defects in new FAZ formation, which blocks the recruitment of the CIFs and other cytokinetic proteins as a secondary, or even tertiary effect. Similarly, although some of the RIT-seq hits appear to have blocked cytokinesis, their absence may have perturbed a much earlier event. I think this is an important point that should be mentioned- that cytokinesis defects can function as a "catch-basin" for many other upstream defects, especially when your only readout is DNA content.

R3.3: We have now commented on this point, adding "due to defects that occur during cytokinesis itself or earlier in the cell cycle in some cases" (see 2nd paragraph in the Discussion).

3.4: It seems strange to use the exocyst as negative control for a cell division screen, considering that it is heavily involved in the later stages of cytokinesis in mammals and yeasts. I realize the authors have looked at exocyst components in *T. brucei* in different work, but I don't know if has been shown conclusively that the *T. brucei* exocyst isn't involved in cell division.

R3.4: Thank-you for pointing to evidence that the exocyst plays a role in cell cycle progression in other cell types. We now use the term 'control cohort' rather than 'negative control cohort' immediately preceding the text; "since none of the exocyst components registered enrichment in the >4C pool, nor in any other experimental pool analysed here" (see 1st paragraph in the 'Cytokinesis ...' Results section). i.e. the exocyst was selected to serve as a control based on the current dataset. Our colleague, Mark Field's team, rather than us, studied the *T. brucei* exocyst previously. We also show data for "further controls that do not appear to have substantial impacts on cell cycle progression, including the mitochondrial RNA editing accessory complex MRB1⁵⁸ and the mitochondrial ATP synthase complex V⁵⁹" (see final paragraph in the 'A profile ...' Results section).

3.5: It should be noted that BSFs appear to be able to fail to complete cytokinesis, restart the cell cycle, and complete the process from an 8C state (Wheeler Mol Micro 2013). This might be worth mentioning- perhaps some of the hits in the screen move cells towards this cell division mode rather than the more conventional one.

R3.5: We do state that “polyploid [$>4C$] cells arise due to endoreduplication, additional rounds of DNA replication without cytokinesis” (see 1st paragraph in the Results section) and have now added “with prominent peaks detected representing 8C and 16C cells” to describe the data in Fig. 7d (see final Results section).

More minor concerns:

3.6: Figure 1A: Could the authors include a schematic branching off the $>4C$ category representing the phenotype described in Figure 8? A cell without multiple furrows but an enlarged nucleus would make the point well.

R3.6: We describe both possible outcomes in the text “additional rounds of DNA replication without cytokinesis, either with ²⁴ or without ^{25,26} mitosis, yielding cells with multiple nuclei or with polyploid nuclei, respectively” (see 1st paragraph in the Results section), but prefer to show only the most commonly observed $>4C$ phenotype here. We have added “(multiple nuclei)” to the Fig. 1a legend to clarify.

3.7: Line 108 : “required G1S progression” should read “required [for] G1S progression”

R3.7: Corrected as suggested.

3.8: Line 155: could you sort zoids, or did you observe them? Were there many cells outside of the largest or smallest gates?

R3.8: Please see R3.2 above.

3.9: Line 258: Exocyst is essential during the latest stages of cell division in other organisms.

R3.9: Please see R3.4 above.

3.10: Line 308: It might be worth mentioning specifically that flagellar beat has been implicated in the last stages of cytokinesis- the trypanin work by Kent Hill is cited but I think you could make a stronger case for why the motile flagellum is important.

R3.10: We do state that “Endoreduplication defects were previously observed following knockdown of ... flagellar proteins ^{7,35}; consistent with the view that flagellar beat is required for cytokinesis in bloodstream form *T. brucei*” (see 1st paragraph in the ‘Cytokinesis ...’ Results section). References 7 and 35 are from the Keith Gull and Kent Hill teams, respectively.

3.11: Line 321: It would be good to have more information about the $<2C$ cells. I realize this is a small set of genes- but is there any unifying theme in terms of function or localization? DOT1 does not appear to have the orange shading expected in the supplemental data, unless I misunderstand how the data is supposed to be presented.

R3.11: We’ve edited the text in the first paragraph of the ‘Validation ...’ Results section and now state “Indeed, other ‘ $<2C$ hits’ mostly encode small hypothetical proteins, seven of which are $73\pm 11\%$ shorter than the average, consistent with low read-count and under-sampling for these hits (Supplementary Fig. 2c). The remaining two hits are a histone chaperone (ASF1B) and a glycolytic enzyme (PFK)”; Supplementary Fig. 2c is also new. We also now state in the text accompanying the online tool that “These coverage plots can be used to assess the number of different RNAi target fragments (that support a given phenotype) for each gene (also considering the untranslated regions), by counting the number of paired stacks of barcoded reads flanking each fragment. For example, the 300 bp 10.10030 CDS appears to be targeted by a single fragment, while the average-length 1.6 kbp 9.12810 CDS appears to be targeted by seven or more fragments”. In relation to DOT1A (Tb927.8.1920), this gene does have orange shading in Additional file 1 and in the original and new online tool (<https://tryp-cycle.pages.dev/> - also see Fig. 2b).

Reviewer comments , second round review -

Reviewer #2 (Remarks to the Author):

The authors have addressed most of points raised in my review, as well as the other reviewers. The issue raised by myself and another reviewer is the issue of confidence levels and how the analysis and visualisation tools sort the genes into classes.

They have modified this tool and it is improved; the nature of such data is that the analysis will always give a result. One has to accept that the significance of this result will only come if the list produced here is successfully informative of future work. As the author's say - that is for future work.

Reviewer #3 (Remarks to the Author):

The updated manuscript has addressed all of my concerns.

22nd August 2022

Marques *et al.*, Genome-scale RNA interference profiling of *Trypanosoma brucei* cell cycle progression defects

Responses to Reviewer Comments:

Reviewer #2:

The authors have addressed most of points raised in my review, as well as the other reviewers. The issue raised by myself and another reviewer is the issue of confidence levels and how the analysis and visualisation tools sort the genes into classes.

They have modified this tool and it is improved; the nature of such data is that the analysis will always give a result. One has to accept that the significance of this result will only come if the list produced here is successfully informative of future work. As the author's say - that is for future work.

We were pleased to see that reviewer 2 agrees that our analysis and visualisation tool is improved.

Reviewer #3:

The updated manuscript has addressed all of my concerns.

We were pleased to see that our updated manuscript has addresses all of reviewer 3's concerns.